**RESEARCH**

# Local CpG density affects the trajectory and variance of age-associated DNA methylation changes

Jonathan Higham[1], Lyndsay Kerr[1], Qian Zhang[2,3], Rosie M. Walker[4,5], Sarah E. Harris[6], David M. Howard[7,8], Emma L. Hawkins[8], Anca-Larisa Sandu[9], J. Douglas Steele[10], Gordon D. Waiter[9], Alison D. Murray[9], Kathryn L. Evans[4], Andrew M. McIntosh[8], Peter M. Visscher[2], Ian J. Deary[6], Simon R. Cox[6] and Duncan Sproul[1,11]*

*Correspondence:
d.sproul@ed.ac.uk

[1] MRC Human Genetics Unit,
Institute of Genetics and Cancer,
University of Edinburgh,
Edinburgh, UK
Full list of author information is
available at the end of the article

## Abstract

**Background:** DNA methylation is an epigenetic mark associated with the repression of gene promoters. Its pattern in the genome is disrupted with age and these changes can be used to statistically predict age with epigenetic clocks. Altered rates of aging inferred from these clocks are observed in human disease. However, the molecular mechanisms underpinning age-associated DNA methylation changes remain unknown. Local DNA sequence can program steady-state DNA methylation levels, but how it influences age-associated methylation changes is unknown.

**Results:** We analyze longitudinal human DNA methylation trajectories at 345,895 CpGs from 600 individuals aged between 67 and 80 to understand the factors responsible for age-associated epigenetic changes at individual CpGs. We show that changes in methylation with age occur at 182,760 loci largely independently of variation in cell type proportions. These changes are especially apparent at 8322 low CpG density loci. Using SNP data from the same individuals, we demonstrate that methylation trajectories are affected by local sequence polymorphisms at 1487 low CpG density loci. More generally, we find that low CpG density regions are particularly prone to change and do so variably between individuals in people aged over 65. This differs from the behavior of these regions in younger individuals where they predominantly lose methylation.

**Conclusions:** Our results, which we reproduce in two independent groups of individuals, demonstrate that local DNA sequence influences age-associated DNA methylation changes in humans in vivo. We suggest that this occurs because interactions between CpGs reinforce maintenance of methylation patterns in CpG dense regions.

## Introduction

DNA methylation is the most common DNA modification found in mammals. It is considered a repressive epigenetic mark at gene promoters and is observed predominantly at cytosines in CpG dinucleotides [1]. In mammals, it is largely erased from the somatic genome in early development and re-established later by the de novo DNA methyltransferases 3A and 3B (DNMT3A and DNMT3B) [2]. This wave of de novo methylation results in a pervasive methylation landscape where 70–80% of CpGs are methylated in most human tissues [3]. Short regions lacking DNA methylation often correspond to promoters and other regulatory elements particularly CpG islands [1]. Subsequently, this DNA methylation pattern is largely maintained by the action of the maintenance DNA methyltransferase, DNMT1 [4].

Despite maintenance methylation, the DNA methylation landscape alters with age. Overall, the DNA methylation content of the genome reduces with age [5] but individual loci gain methylation. Changes at some loci are sufficiently reproducible across individuals as to enable the statistical derivation of accurate predictors of age termed epigenetic clocks [6, 7]. Accelerated aging inferred using epigenetic clocks has been observed in human diseases and health-associated conditions such as obesity [8]. It is also predictive of the development of disease [8] and mortality [9]. Similar epigenetic clocks have been derived for other mammalian species [10–13]. In mice, interventions associated with increased lifespan also associate with decreases in epigenetic age measured by murine epigenetic clocks [10, 12]. In addition to epigenetic clocks, DNA methylation changes that are inconsistent between individuals have also been described and termed epigenetic drift [14]. For example, increased divergence in the DNA methylation patterns of twins is observed with age [15].

The mechanisms underpinning these changes in DNA methylation with age remain unclear. Losses of DNA methylation with age are thought to occur primarily in heterochromatic, late replicating genomic regions [16]. Conversely, DNA methylation gains have been associated with CGIs that are targeted by polycomb repressive complexes in embryonic stem cells [17–19]. Although epigenetic clocks are designed as sparse predictors of age and therefore are likely to capture a range of diverse processes ongoing in cells and tissues [20], epigenetic clock loci have been observed to include enhancers [10] suggesting that changes in the methylation level at enhancers may occur with age. The majority of variation in DNA methylation seen between cell types also occurs at enhancers [21]. An analysis of age-associated epigenetic changes has suggested that many correlate with variation in the proportions of different cell types in the blood [22]. Furthermore, some epigenetic clocks have been suggested to capture changes in the proportions of cell types because they correlate with cell type proportions [8]. Age-associated variable loci have also been reported to occur in regions of the genome regulated by polycomb repressive complexes [23].

Analysis of steady-state DNA methylation patterns in human populations has found differences in the methylation levels of individual loci between people that associate with sequence polymorphisms. These have been characterized as allele-specific methylation or methylation quantitative trait loci (meth-QTLs) [24, 25]. This suggests that DNA sequence can program local DNA methylation levels, a hypothesis supported by the inheritance pattern of allele-specific methylation in families [26] and analysis

of the methylation patterns of ectopic DNA sequences integrated into cell lines [27]. Whether DNA sequence plays a role in age-associated changes in DNA methylation is less clear. Some studies have provided evidence that genetic variants affect how DNA methylation changes with age at individual loci [28, 29]. However, a study of mice possessing a copy of human chromosome 21 suggested that local sequence plays a little role in determining the rate of age-associated DNA methylation changes [30]. In this study, the human chromosome accumulated age-associated changes at a similar rate to the mouse genome rather than the rate observed in its native human context.

Here we examine longitudinal DNA methylation patterns at 345,895 individual CpGs in blood DNA samples from people aged between 67 and 80 to determine which loci show changes in DNA methylation level and provide insight into mechanisms that might underpin age-associated epigenetic changes, particularly in later life. We demonstrate that the methylation patterns of low CpG density regions are more likely to change with age and do so variably in later life.

## Results

### Longitudinal methylation trajectories reveal changes at individual epigenetic clock loci

In order to understand the factors that are responsible for age-associated alterations in DNA methylation at the CpG level, we analyzed longitudinal DNA methylation data collected from blood samples taken from the Lothian Birth Cohort of 1936 (LBC) [31–33]. This cohort consists of 1091 individuals, whose blood was assayed at multiple timepoints on Illumina Infinium 450k arrays between the ages of 67 and 80 (see Table 1). To robustly quantify DNA methylation alterations with age, we focused on the 600 individuals for whom 3 or more datapoints were available and 345,895 reliably measured autosomal CpG probes whose signal is not directly affected by SNPs or cross-hybridisation [34].

We modelled methylation trajectories for each CpG and individual as linear models of the Infinium beta values with age (example shown in Fig. 1a). This approach estimates every individual's slope independently and can account for heterogenous groups within the data. Mixed effect models provide an alternative approach [35]. However, mixed effects models borrow information between individuals implicitly assuming all individuals belong to a single group. Our mean rates of change derived from individual trajectories were highly correlated with the rates of change estimated from a mixed effects model including a random intercept (Additional file 1: Fig S1a, Pearson's $R = 0.999$, $p < 2.2 \times 10^{-16}$ for the 345,890 CpGs that could be modelled in this manner).

**Table 1** Demographics of LBC participants used in this study. The mean age in years at each measurement are indicated along with the range. The number of observations at each measurement is also indicated. In total, 351 individuals had data for all 4 measurements and 249 for three of the measurements

|                    | 1st measurement | 2nd measurement | 3rd measurement | 4th measurement |
| ------------------ | --------------- | --------------- | --------------- | --------------- |
| Mean age (min/max) | 69.6 (67.7, 71,3) | 72.6 (71.0, 74.2) | 76.3 (74.7, 77.7) | 79.3 (78.0, 80.9) |
| No. females        | 265             | 271             | 259             | 224             |
| No. males          | 295             | 307             | 292             | 238             |

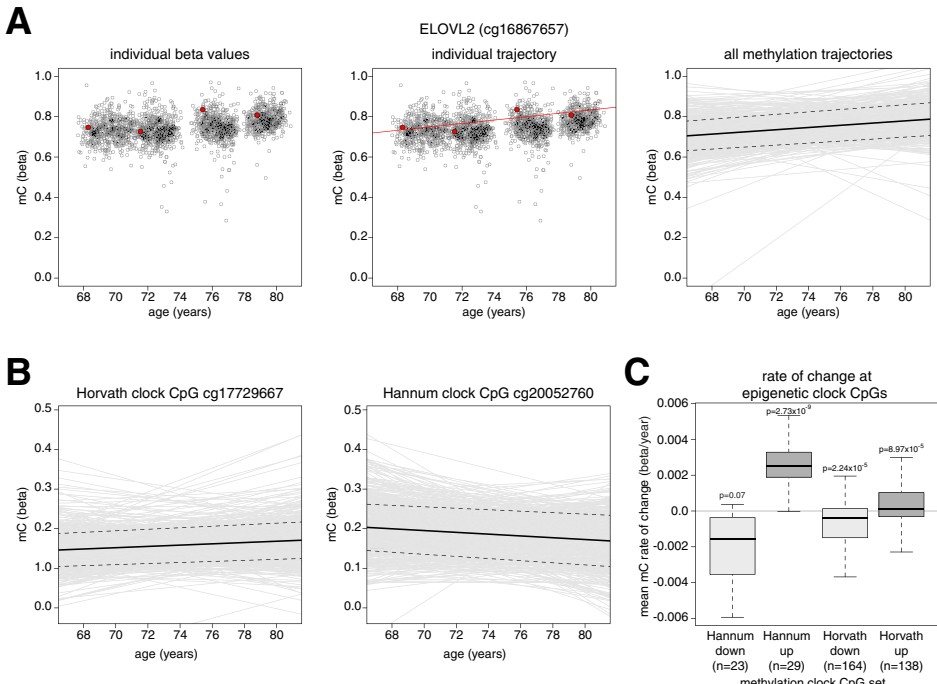

**Fig. 1** Longitudinal methylation trajectories reveal changes at individual epigenetic clock loci. **a** ELOLV2 shows increases in methylation across individuals. Plots of methylation levels at ELOVL2 CpG cg16867657 showing data from one individual in the cohort and their methylation trajectory (left and middle panel, red points and line), and methylation trajectories for all individuals (right panel, gray lines). In the right panel, the bold line indicates the mean methylation trajectory, and the dashed lines are the 95% confidence intervals. **b** Examples of methylation trajectories observed for epigenetic clock CpGs. Individual methylation trajectories are indicated by gray lines. The mean methylation trajectory is indicated by the bold line and the dashed lines are the 95% confidence intervals. **c** Methylation trajectories recapitulate the predicted behavior of epigenetic clock CpGs. Boxplots showing the calculated mean rates of change for CpGs that are part of the Hannum or Horvath epigenetic clocks split by their reported direction of change. *P*-values were calculated using *T*-test. Lines = median; Box = 25th–75th percentile; whiskers = 1.5 × interquartile range from box; n indicates the number of CpGs in each group

To test whether age-associated DNA methylation changes at individual loci could be measured using our approach, we first examined the CG probe *cg16867757* in the *ELOV2L* promoter which shows strong age-associated changes in DNA methylation [7, 36]. We observed a highly significant gain of methylation with age at the locus in the LBC cohort, validating our approach (Fig. 1a, T-test, $p < 2.2 \times 10^{-16}$). To further test whether we could reliably estimate changes at individual loci, we next examined epigenetic clock loci. Epigenetic clocks have been defined as statistical instruments whose output is strongly correlated with age [20]. Individual clock loci show DNA methylation levels that correlate with age in cross-sectional analyses [12]. Given their use in predicting age, they would be expected to show consistent trajectories between individuals. We focused on the behavior of CpGs that are included in the widely used Hannum and Horvath epigenetic clocks [6, 7]. 88% of the CpGs in the Hannum epigenetic clock and 80% of the CpGs in the Horvath epigenetic clock had a statistically significant change with age in the direction predicted by the original studies (46 out of 52 and 241 out of 302 respectively, $p < 0.05$, *T*-test, examples in Fig. 1b and aggregate analysis Fig. 1c) [6, 7]. The slopes of the CpGs making up these epigenetic clocks in LBC were also highly

consistently correlated with those calculated from a cross-sectional cohort of 5101 individuals from the Generation Scotland study who had their DNA methylation levels profiled on Illumina EPIC arrays (Additional file 1: Fig S1b) [37, 38]. The absolute rate of change of clock CpGs were modest compared to other CpGs whose methylation levels changed significantly with age in accordance with previous observations of individual clock loci in cross-sectional analyses (Additional file 1: Fig S1c) [12].

These results suggest that the analysis of individual methylation trajectories in the LBC cohort can detect previously described age-associated changes in DNA methylation at individual CpGs.

### A large number of CpGs show age-associated changes in DNA methylation levels in later life

Having demonstrated that we can measure predicted changes in DNA methylation with age using individual methylation trajectories, we then examined the rates of change at individual CpGs across the genome to understand which loci might show changes in DNA methylation levels with age in LBC.

Of the 345,895 CpGs in the dataset, 182,760 (52%) show significant changes in DNA methylation levels with age in the LBC blood samples (Bonferroni-corrected $p < 0.01$, $T$-test of individual linear model slopes). Many changes in DNA methylation with age have been attributed to changes in the proportions of cell types in the blood [22]. To understand the degree to which variation in cell type proportions might explain our observations, we made use of directly measured neutrophil, lymphocyte, monocyte, eosinophil and basophil counts from the same blood samples. We estimated rates of change for the 5 measured cell types by fitting a linear model to the counts in each individual. We then modelled the rate of change in DNA methylation with age as a linear function of these cell type rates across individuals for each of the 182,760 significantly changing CpGs. The rate of change in DNA methylation at 17,862 CpG sites was significantly explained by these models (9.77%, F-tests, Bonferroni-corrected $p < 0.01$). At these sites, the proportion of variation in rate of change of DNA methylation with age explained by variation in the blood cell count rates was low (mean $R^2 = 0.150$, Additional file 1: Fig S2a). This suggests that relatively few of the age-associated methylation changes we observe were caused by changes in the proportions of cell types in the blood with age. Estimates of cell type proportions can also be derived using cell-type-specific DNA methylation differences [39]. We applied the Houseman algorithm to estimate the proportions of granulocytes, B-cells, CD4 T-cells, CD8 T-cells, natural killer cells, and monocytes present in each DNA methylation sample before repeating our analysis. Using these estimated white blood cell counts, 118,059 of the age-associated CpGs showed variation that was significantly explained by rates of change in cell counts (64.60%, F-tests, Bonferroni-corrected $p < 0.01$). However, again the proportion of variance explained by the variation in blood cell rates of change was modest (mean $R^2 = 0.197$, Additional file 1: Fig S2b).

Taken together, these analyses suggest that a large number of CpGs show age-associated changes in DNA methylation levels in LBC. While some of these changes are partially explained by changes in the proportions of cell types within the blood, variation in

the blood cell counts explains a small proportion of the variation in DNA methylation levels observed at most of these CpGs.

### A subset of CpGs gain methylation in later life

In order to understand what else might explain the age-associated changes in DNA methylation levels we observed in the LBC blood samples, we then examined the distribution of mean slopes for the CpGs showing a significant change in DNA methylation with age.

This distribution was significantly skewed towards loci gaining DNA methylation (Fig. 2a, $p < 2.2 \times 10^{-16}$ by *T*-test of mean linear model slopes). A distinct shoulder of CpGs with more rapid gains of methylation was also apparent on the histogram. We defined these rapidly gaining CpGs as those with a rate of methylation change > 0.016 beta per year (>0.16% methylation per year, 8322 rapid gain CpGs, Additional file 2: Table S1; example *cg22926528* shown in Fig. 2b). Rapid gain CpGs also showed significantly higher rates of methylation gain than other CpGs when slopes were corrected for measured white blood cell counts from the LBC cohort (Additional file 1: Fig S2c, $p < 2.2 \times 10^{-16}$ Wilcoxon test) suggesting that the observed gains of DNA methylation did not result from altered blood composition with age. They also had significantly higher rates of change when the data were corrected for cell counts estimated from DNA methylation data using the Houseman algorithm (Additional file 1: Fig S2d, $p < 2.2 \times 10^{-16}$ Wilcoxon test).

To understand why these CpGs might gain methylation, we examined where they were located in the genome. Compared to all other CpGs in the dataset, the rapid gain CpGs were significantly depleted from CpG islands and the regions surrounding CGIs which have been termed shores [40] (Fig. 2c). They were instead enriched in the bodies of coding genes (63.3% of CpGs, Fig. 2c) and large genomic regions of reduced methylation termed partially methylated domains (PMDs) [41] defined across 40 tumor and 9 normal samples (34.6% of CpGs, Fig. 2c) [16]. PMDs are known to be heterochromatic, gene poor and have a lower CpG density than other regions of the genome [16, 41, 42]. Consistent with their enrichment in PMDs, the regions surrounding rapid gain CpGs had a significantly lower CpG density than other CpGs analyzed (Additional file 1: Fig S2e, Wilcoxon test $p < 2.2 \times 10^{-16}$).

To further understand the characteristics of this set of CpGs, we cross-referenced them to chromatin state data generated by the ENCODE and Roadmap Epigenomic projects [43, 44]. These projects have used hidden Markov models to partition the genome into distinct chromatin states (ChromHMM) [45]. Consistent with their observed enrichment in gene bodies and PMDs, the rapid gain CpGs were significantly enriched in ENCODE-defined transcriptional and heterochromatic chromatin states in GM12878 lymphoblastoid cells (11.0% and 57.7% of CpGs with transcriptional and heterochromatin annotations respectively, Fig. 2d)[36]. Similarly, they were most enriched in the heterochromatin-associated quiescent state across a set of 23 primary blood cell types whose chromatin states were defined by the Roadmap Epigenomics project (Additional file 1: Fig S2f)[37]. Transcriptional states were also enriched in these primary blood cells but to a lesser degree (Additional file 1: Fig S2f).

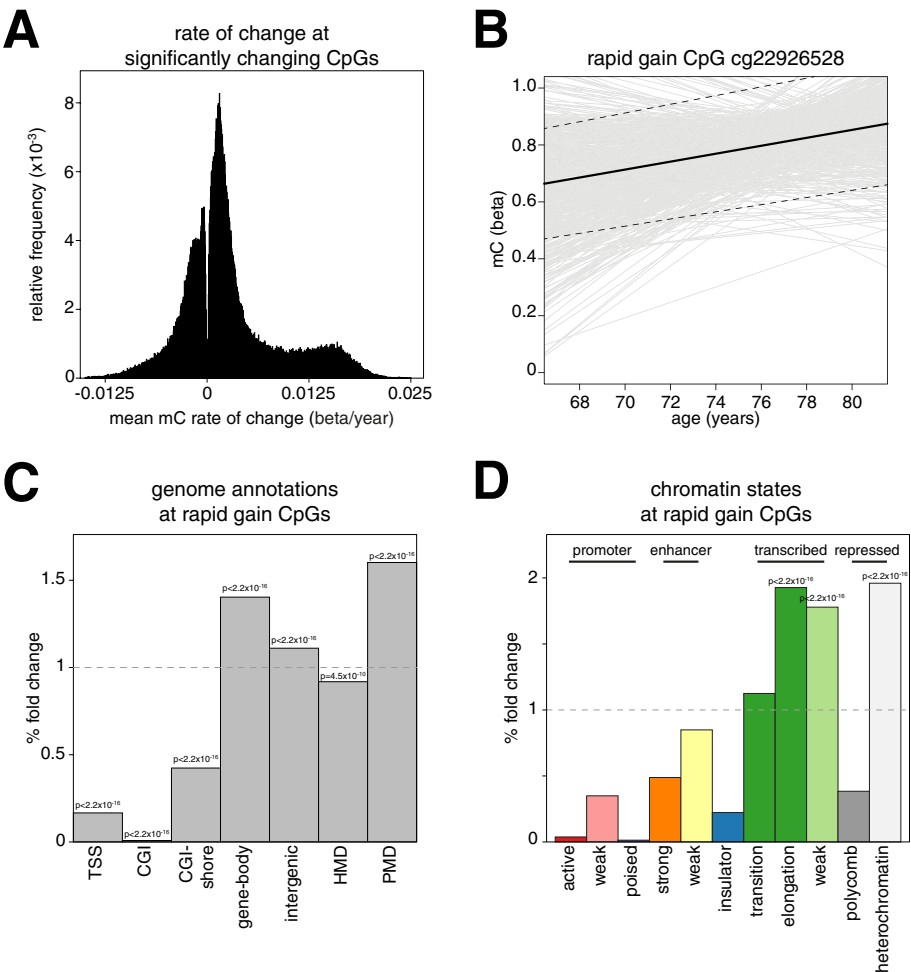

**Fig. 2** A subset of CpGs gain methylation in later life. **a** A subset of CpGs gain methylation in LBC. Histogram of the mean methylation trajectories for the 182,760 CpGs whose slope significantly deviates from 0 (*T*-test, Bonferroni-corrected *p* < 0.01). **b** Example of a rapid gain CpG. Individual methylation trajectories are indicated by gray lines. The mean methylation trajectory is indicated by the bold line and the dashed lines are the 95% confidence intervals. **c** Rapid gain CpGs are depleted from CGIs and enriched in gene bodies and intergenic regions. Barplot showing the % fold change observed for rapid gain CpGs in different genome annotations versus the background of all analyzed CpGs. *P*-values are from 2-sided Fisher's exact tests. PMDs = partially methylated domains; HMDs = highly methylated domains. **d** Rapid gain CpGs are enriched in transcription and heterochromatin states in GM12878 cells. Barplot showing the % fold change observed for rapid gain CpGs in different chromatin states in GM12878 cells versus the background of all analyzed CpGs. Shown are significant *P*-values from 1-sided Fisher's exact tests

Our analyses therefore suggest that changes in methylation in the LBC cohort are most apparent at a subset of CpGs, largely located in heterochromatic, low CpG density regions.

### Local SNPs associate with altered CpG methylation trajectories

Having uncovered a set of CpGs which rapidly gained methylation with age, we then asked what factors led to differences in the trajectories of DNA methylation alterations between CpGs. Previous reports suggest a potential influence of genetic variation on

age-associated DNA methylation changes [28, 29] so we tested for associations between local (*cis*; within 1 Mb) SNPs and the rate of change of DNA methylation at individual CpGs with age across the cohort.

Our analysis uncovered 4673 slope-Quantitative Trait Loci (slope-QTLs) representing 1456 unique CpGs (examples in Fig. 3a). The linkage disequilibrium structure of the human genome means that linked SNPs would be expected to be associated with the same CpG. In order to determine the number of independent associations, we used conditional analysis to resolve these slope-QTLs into 1487 SNP-CpG pairs designating the closest independent SNP in each case as the lead SNP (Additional file 2: Table S2). Only 31 CpGs were independently associated with more than 1 SNP. Each lead SNP was significantly associated with the rate of change of methylation at a mean of 1.06 CpGs (range 1 to 16, Additional file 2: Table S2). We validated these associations using another method by testing for an age $\times$ genotype interaction effect in a standard linear model. Nearly all of these SNP-CpG pairs (1334, 90.3%) had a significant interaction (Bonferroni-corrected, $p < 0.05$) and the effect size determined from the slope of the individual linear models, and the effect size of the age $\times$ genotype interaction within the population were significantly correlated (Spearman Rho $= 0.977$, $p < 2.2 \times 10^{-16}$, Additional file 1: Fig S3a). All slope-QTL CpG-SNP pairs were also significant when re-analyzed using beta values that had been corrected for variation in the proportions of different blood cell types both when directly counted or estimated from DNA methylation data using the Houseman algorithm (Additional file 2: Table S2). We found that 93 (6.66%) of the slope-QTL SNPs were present in the NHGRI-EBI GWAS catalog (Additional file 2: Table S4) [46]. The most frequent annotation was for SNPs associated with "Hip circumference adjusted for BMI" (8 SNPs). In addition, 20 of the SNPs were annotated as being previously associated with blood cell traits such as "platelet count" and "eosinophil counts" (for full list see "Materials and methods").

We also tested for *trans*-slope-QTLs independent of SNP-CpG genomic distance. This analysis uncovered 9 significant SNP-CpG pairs (Additional file 2: Table S5). The low number was likely due to the multiple testing burden associated with testing every SNP against every CpG. Given the low number of observed *trans*-slope-QTLs, we focused on the analysis of *cis*-slope-QTLs.

Although we had set a threshold of 1 Mb when uncovering *cis*-slope-QTL, the lead SNPs were located close to the slope-QTL CpGs (Fig. 3b). At 53% of the slope-QTLs, the lead SNP and CpG were within 1 kb of each other. Whereas most SNPs (1346/1397, 96.3%) were only associated with a single CpG at the genome-wide Benjamini-Hochberg-corrected significance FDR $< 0.05$, we observed that other CpGs close to those in

(See figure on next page.)

**Fig. 3** Local SNPs associate with altered CpG methylation trajectories. **a** Examples of slope-QTLs. Spaghetti plots and boxplots of 3 slope-QTL CpG-SNP pairs. Left, spaghetti plots of individual methylation trajectories separated by genotype. Thin lines represent individual methylation trajectories and thick lines the mean methylation trajectory for that genotype. Right, boxplots of slope separated by genotype. Lines = median; Box = 25th–75th percentile; whiskers = 1.5 × interquartile range from box. SNP genotypes are annotated relative to the forward strand. **b** Slope-QTL SNPs are located in close proximity to the CpGs they affect. Histogram of the distances between slope-QTL lead SNPs and the CpGs they are paired with. **c** Nearby CpGs are also affected by slope-QTL SNPs. Line plot of the effect sizes calculated for CpGs within −/+ 1 Kb of slope-QTL CpGs using each slope-QTL's lead SNPs. Plotted is the mean normalized effect size in 50 bp Windows. Bold lines show the mean effect size and dashed lines and shaded area show the 95% confidence intervals. The data are shown in red and the results of 1000 random permutations shown in black

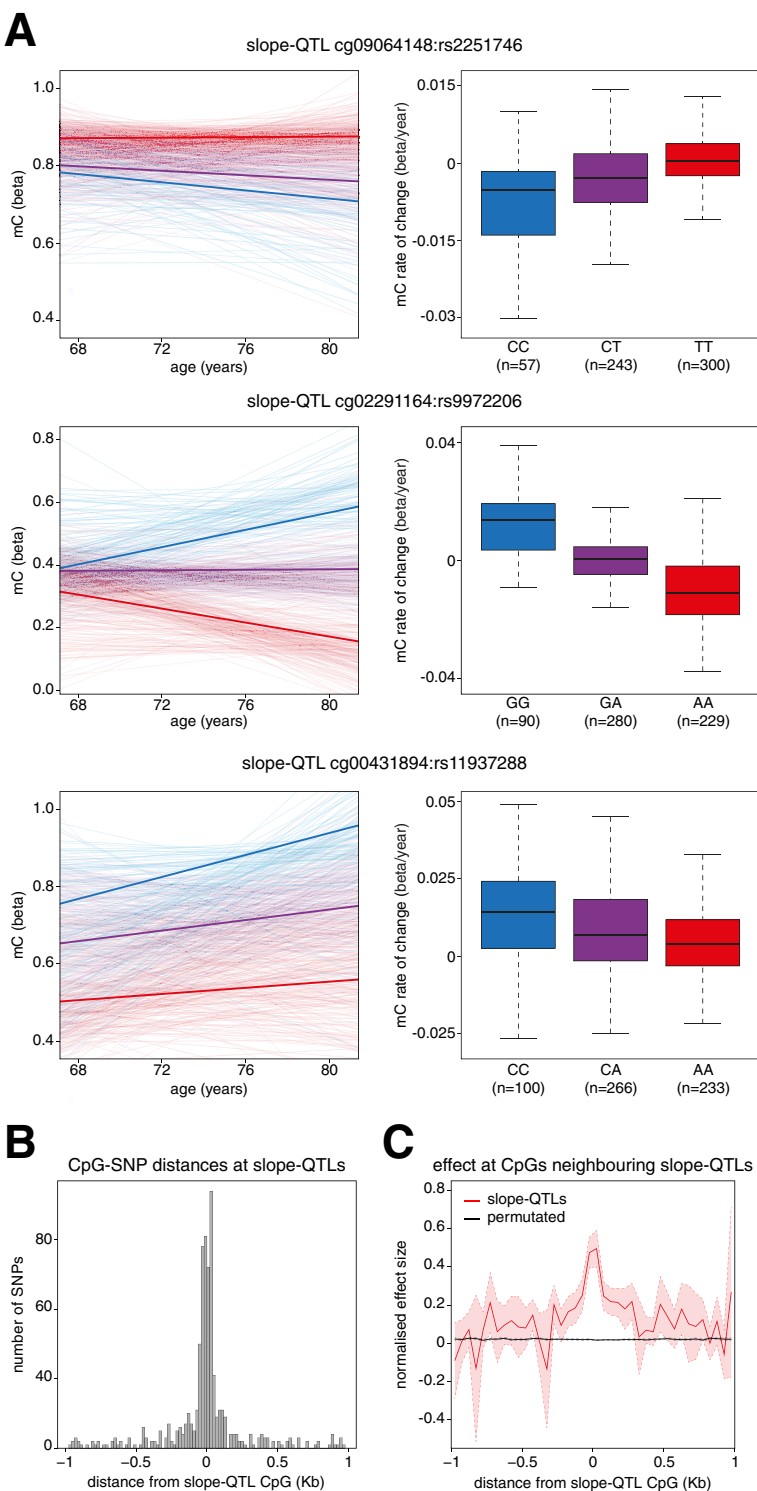

**Fig. 3** (See legend on previous page.)

slope-QTLs showed correlated effects that were below the multiple testing-corrected significance threshold (Fig. 3c). This strongly suggests that these slope-QTLs are driven by specific effects of genotype on methylation change with age within local genomic

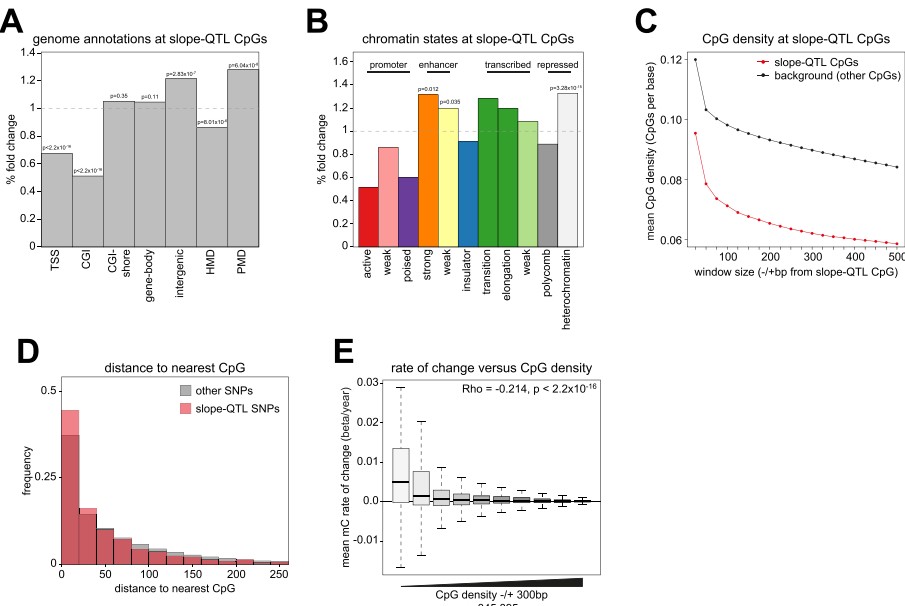

**Fig. 4** Local CpG density affects age-associated DNA methylation trajectories. **a** Slope-QTL CpGs are enriched in intergenic regions. Barplot showing the % fold change observed for slope-QTL CpGs in different genome annotations versus the background of all analyzed CpGs. *P*-values are from 2-sided Fisher's exact tests. **b** Slope-QTL CpGs are enriched in enhancer and heterochromatin states in GM12878 cells. Barplot showing the % fold change observed for slope-QTL CpGs in different chromatin states in GM12878 cells versus the background of all analyzed CpGs. Shown are significant *P*-values from 1-sided Fisher's exact tests. **c** Slope-QTL CpGs are found in regions of low CpG density. Line plot showing the mean CpG density around slope-QTL CpGs in different window sizes. Red shows slope-QTL CpGs and black shows all other CpGs assayed. **d** Slope-QTL SNPs are found close to CpG sites. Histogram of the distances between slope-QTL SNPs and their nearest CpG site. Red shows the distribution for slope-QTL SNPs and gray shows all other SNPs assayed. **e** CpG density is a major determinant of variation in methylation trajectories with age. Boxplot showing mean methylation trajectories plotted against CpG density ± 300 bp from the CpG. The Spearman correlation, Rho, and *p*-value, *T*-test, for the association are given. For plotting, CpG density is binned into equally sized groups. Lines = median; Box = 25th–75th percentile; whiskers = 1.5 × interquartile range from box

regions. It also makes it unlikely that slope-QTLs are caused by the disruption of a single CpG probe by a SNP linked to the lead SNP as the rate of change at multiple CpGs was associated with the lead SNP in many cases.

Overall, our analyses suggest that local SNPs can affect the rate at which DNA methylation changes with age at CpGs located in their vicinity.

### Slope-QTLs are located in CpG-poor regions of the genome

To understand the mechanistic basis of slope-QTLs, we analyzed their genomic locations. CpGs that were part of slope-QTLs were significantly depleted from CpG islands and their shores but significantly enriched in intergenic regions and PMDs (35.4% and 27.6% of CpGs respectively, Fig. 4a). Consistent with this, slope-QTL CpGs were also significantly enriched in chromatin states associated with enhancers and heterochromatin in GM12878 lymphoblastoid cells (12.3% and 39.1% of CpGs respectively, Fig. 4b). Significant enrichments in enhancer and quiescent heterochromatin states were also seen in 95.7% and 100% of the 23 primary blood cell types analyzed in the Roadmap Epigenomics project (*p* < 0.05, Additional file 1: Fig S4a).

*Cis*-meth-QTLs have previously been shown to be enriched at enhancers and associated with SNPs altering local transcription factor (TF) binding sites [24, 25]. We therefore asked whether this might also be the case for slope-QTLs. We compared the locations of slope-QTL CpGs to binding sites defined for 111 TFs in GM12878 cells [43]. Only a single TF, NFE2, was significantly enriched at slope-QTL CpGs and this was only observed at 6 loci (Additional file 2: Table S6). The vast majority of TF binding sites analyzed were instead significantly depleted from slope-QTLs (90, 84.91%, Additional file 2: Table S6). We similarly analyzed the lead SNPs associated with slope-QTLs to ask if they overlapped TF binding sites in GM12878. In this analysis, the binding sites for 14 TFs were significantly enriched at the SNPs (Additional file 2: Table S7). However, even the most significantly enriched, CTCF, was only observed at 4.44% of the slope-QTL SNPs. Overall, this suggests that unlike *cis*-meth-QTLs, the majority of *cis*-slope-QTLs are not associated with TF binding sites.

Previous work has highlighted local CpG density and the sequence surrounding intergenic CpGs as being associated with their methylation levels [16, 47]. Given the enrichments, we observed in intergenic annotations for slope-QTL CpGs and the observation that rapid gain CpGs have a low surrounding CpG density, we wondered if slope-QTLs might also be located in regions of low CpG density. Slope-QTL CpGs were located in regions with a significantly lower local genome CpG density than other CpGs assayed on the Infinium array (Fig. 4c). The difference persisted across different window sizes surrounding the slope-QTL CpG although the effect was strongest around $\pm 325$ bp (Fig. 4c, Additional file 1: Fig S4b). We then asked whether the SNPs associated with alterations in methylation trajectories at slope-QTLs might affect the local sequence composition around CpGs. To do so, we measured the distance between the slope-QTL lead SNP and its nearest genomic CpG (irrespective of whether it is the Infinium array assayed slope-QTL CpG). Despite being located in CpG-poor regions, slope-QTL lead SNPs were also found significantly closer to their closest CpG than non-slope-QTL SNPs in our analysis (*T*-test, $p = 5.4 \times 10^{-12}$) and 11.4% directly affected a CpG site or the bases adjacent to one (159 out 1397).

This suggests that slope-QTLs are found within low CpG density regions and SNPs associated with them frequently alter the regions CpG density or the bases neighboring CpGs.

### Local CpG density affects age-associated DNA methylation trajectories

These results suggest that alterations in the local sequence context around CpGs affect their methylation trajectory, particularly in regions of low CpG density. To understand the relationship between methylation trajectories and CpG density more generally, we analyzed how methylation trajectories varied with local CpG density genome-wide. We observed that the magnitude of change at CpGs with age was significantly negatively correlated with their local CpG density (Rho $= -0.214$, $p < 2.2 \times 10^{-16}$, *T*-test, Fig. 4e). CpGs with a lower surrounding CpG density were more likely to have altered methylation levels than those in higher CpG density regions, and this was skewed towards gains of methylation. The relationship between mean methylation trajectories and CpG density was also observed when we corrected for measured and Houseman estimated white blood cell counts from the LBC cohort (Rho $= -0.218$ and $-0.237$ respectively,

both $p < 2.2 \times 10^{-16}$ T-tests, Additional file 1: Fig S4c, d respectively) demonstrating that the relationship between CpG density and age-associated methylation changes did not result from altered blood composition with age. The strength of the association was also similar when we removed CpG probes located in CpG islands (Rho $= -0.213$, $p < 2.2 \times 10^{-16}$, *T*-test, Additional file 1: Fig S4e), suggesting it was driven by CpGs lying outside of CpG islands.

While the mean trajectories of CpGs located in low CpG density regions had a median gain of methylation with age, mean trajectories were also far more variable in low CpG density regions (Fig. 4e). The squared residuals of the model fitted between mean slope and CpG density were significantly negatively correlated with CpG density (Spearman's Rho $= -0.561$, $p < 2.2 \times 10^{-16}$), and a Breusch-Pagan test for heteroscedasticity was highly significant ($p < 2.2 \times 10^{-16}$). This confirms that mean methylation trajectories were significantly more variable for low CpG density regions than for high CpG density regions. We wondered whether this variability might also occur between individuals. To test this hypothesis, we calculated the variance in slope across individuals for each CpG. This slope variance displayed a parabolic relationship with the mean level of methylation at CpGs across the timepoints (Additional file 1: Fig S4f). After accounting for this relationship (see "Materials and methods"), we validated whether predicted differences in slope variance between CpGs could be observed. Given their utility in measuring age, CpGs which are part of the Hannum and Horvath epigenetic clocks would be expected to have consistent methylation trajectories and thus low inter-individual variance in slope. We found that this was the case and the slope variance of clock CpGs was significantly lower than other CpGs (Additional file 1: Fig S4g, Wilcoxon test, $p = 8.72 \times 10^{-6}$). Analyzing CpG slope variance between individuals more globally, we found a significant association between slope variance and local CpG density with CpGs in lower density regions having a greater slope variance (Additional file 1: Fig S4h, $p < 2.2 \times 10^{-16}$ by *T*-test).

Our genome-wide analyses therefore suggest that local CpG density affects a CpG methylation trajectory with age and in particular that methylation trajectories in low CpG density regions are more variable than those of high CpG density regions.

### Many age-associated changes in DNA methylation are specific to older individuals

We next sought to determine whether the effect of CpG density of age-associated DNA methylation changes was observed in individuals across a wider range of ages. We therefore examined DNA methylation patterns in 5101 individuals from the Generation Scotland cohort whose blood methylation patterns had been profiled on Illumina EPIC arrays. We first fitted linear models of beta value against age to 758,255 CpGs for the 406 individuals aged > 65 years to match the age range present in LBC. The *p*-values associated with the linear models of the 182,536 CpGs significantly changing in LBC and present on the EPIC array were significantly lower than those for the other CpGs analyzed in the LBC cohort (one-sided Wilcoxon $p < 2.2 \times 10^{-16}$, Additional file 1: Fig S5a). In addition, there was a significant correlation between the estimated slopes of the LBC significantly changing CpGs in the Generation Scotland individuals aged > 65 (Rho $= 0.651$, $p < 2.2 \times 10^{-16}$). However, the *p*-values of LBC significantly changing CpGs

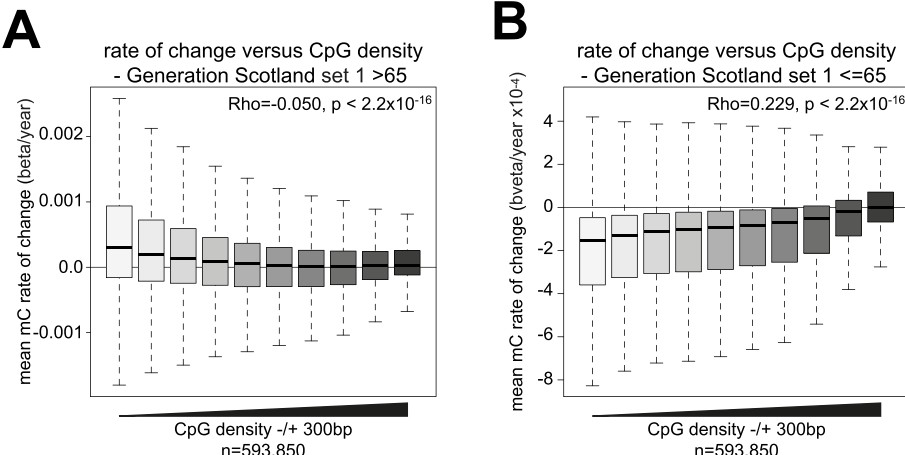

**Fig. 5** Many age-associated changes in DNA methylation are specific to older individuals. **a** CpG density associates with variation in age-associated methylation changes in a second independent cohort. Boxplot showing estimated rates of change in DNA methylation from the 406 individuals aged > 65 in Generation Scotland cohort set 1 plotted against CpG density ± 300 bp from the CpG. The Spearman correlation, Rho, and *p*-value, *T*-test, for the association are given. For plotting, CpG density is binned into equally sized groups. Lines = median; Box = 25th–75th percentile; whiskers = 1.5 × interquartile range from box. **b** CpG density associates with changes in DNA methylation in younger individuals. Boxplot showing estimated rates of change in DNA methylation from the 4695 individuals aged ≤ 65 in Generation Scotland cohort set 1 plotted against CpG density ± 300 bp from the CpG. The Spearman correlation, Rho, and *p*-value, *T*-test, for the association are given. For plotting, CpG density is binned into equally sized groups. Lines = median; Box = 25th–75th percentile; whiskers = 1.5 × interquartile range from box

were not significantly shifted and an inverse correlation in slopes was observed when we analyzed the 4695 individuals whose age was ≤ 65 in Generation Scotland (one-sided Wilcoxon $p = 1.00$, Rho = −254, Additional file 1: Fig S5b). This suggests that similar age-associated changes in DNA methylation were observed in LBC and older individuals from the Generation Scotland cohort, but that these were distinct from DNA methylation changes in younger individuals.

We then asked whether an association between age-associated DNA methylation changes and CpG density was also observed in Generation Scotland. By considering all 593,850 non-CpG island CpGs on the EPIC array, we observed a weak correlation between CpG density and the estimated rate of change of DNA methylation in the members of Generation Scotland aged > 65 (Rho = −0.050, $p < 2.2 \times 10^{-16}$, *T*-test, Fig. 5a). However, we observed greater variation in estimated rates of change at CpGs with lower surrounding CpG densities (Fig. 5a). Consistent with this, there was a significant negative correlation between the squared residuals of a linear model fitted to the estimated rate of change of DNA methylation with age and local CpG density (Rho = −0.228, $p < 2.2 \times 10^{-16}$) and a Breusch-Pagan test for heteroscedasticity was highly significant ($p < 2.2 \times 10^{-16}$). The summed absolute residuals for the models fitted to each CpG against age for individuals > 65 were also significantly negatively correlated with CpG density (Rho = −0.219, $p < 2.2 \times 10^{-16}$, Additional file 1: Fig S5c) with higher residuals observed for CpGs found in low CpG density parts of the genome. This suggests that in Generation Scotland members aged > 65, DNA methylation changes with age were significantly more variable at CpGs in low CpG density parts of the genome than in high CpG density regions. In contrast, we observed a stronger correlation between the

estimated rate of change in DNA methylation and CpG density in individuals aged $\leq 65$ with low CpG density regions more likely to lose methylation with age (Rho $= 0.279$, $p$-value $= 2.2 \times 10^{-16}$, $T$-test, Fig. 5b). Here we also observed significant heteroscedasticity in the residuals ($p < 2.2 \times 10^{-16}$, Breusch-Pagan test) and the summed absolute residuals of the linear models fitted for individual CpGs negatively correlated with CpG density (Rho $= -0.236$, $p < 2.2 \times 10^{-16}$, $T$-test) suggesting the existence of variability in low CpG density regions.

We then asked whether these associations were reproduced in an independent group of 4450 individuals from Generation Scotland (Generation Scotland set 2) who had also been profiled on Illumina EPIC arrays. Estimated rates of change weakly correlated with CpG density in the 519 individuals > 65 from this 2nd set (Rho $= 0.086$, $p < 2.2 \times 10^{-16}$, Additional file 1: Fig S5d). However, like in Generation Scotland set 1, highly significant heteroscedasticity in the squared residuals was observed ($p < 2.2 \times 10^{-16}$, Breusch-Pagan test) and the summed absolute residuals of the linear models fitted to individual CpGs were significantly negatively correlated with CpG density (Rho $= -0.150$, $p < 2.2 \times 10^{-16}$, Additional file 1: Fig S5e). In the 3931 individuals of Generation Scotland set 2 aged $\leq 65$, we observed that CpGs in low CpG density regions were significantly more likely to lose DNA methylation (Rho $= 0.173$, $p < 2.2 \times 10^{-16}$, Additional file 1: Fig S5f) and that there was a significant negative correlation between summed absolute residuals for the linear models fitted to CpGs and CpG density (Rho $= -0.163$, $p$-value $= 2.2 \times 10^{-16}$) replicating observations made in Generation Scotland set 1.

Taken together, our analyses of Generation Scotland suggest that the CpGs in lower CpG density regions have more variable changes in DNA methylation with age in older individuals and in addition are more likely to lose methylation with age in younger individuals.

## Discussion

While alterations in DNA methylation patterns with age have been widely observed in humans and are associated with health, the molecular mechanisms underpinning them remain unclear. Here we use human longitudinal DNA methylation profiles to demonstrate a strong effect of local CpG density on how DNA methylation changes with age (Fig. 6a) and that this can be altered by polymorphisms around CpGs (Fig. 6b).

Previous work has described genetic influences on how DNA methylation patterns change with age. Genome-wide association studies (GWAS) find a number of loci that affect aging as estimated by epigenetic clocks [48–50]. In addition, analyses of rare Mendelian traits suggest that epigenetic aging is accelerated in two growth disorders, Sotos syndrome and Tatton-Rahman-Brown syndrome [51, 52]. These two sets of studies analyzed overall changes in the methylome with age rather than factors influencing the rate of change at individual loci. Further population analyses have also demonstrated the potential for genetic effects on how methylation changes with age at individual loci [28, 29] but did not define the mechanisms that underpin these associations. In contrast, an experiment analyzing DNA methylation in mice possessing a copy of human chromosome 21 found that the introduced human loci changed their methylation status at a rate consistent with the mouse rather than human genome [30]. Based on these observations, the authors suggested that local sequence has little effect in determining the rate

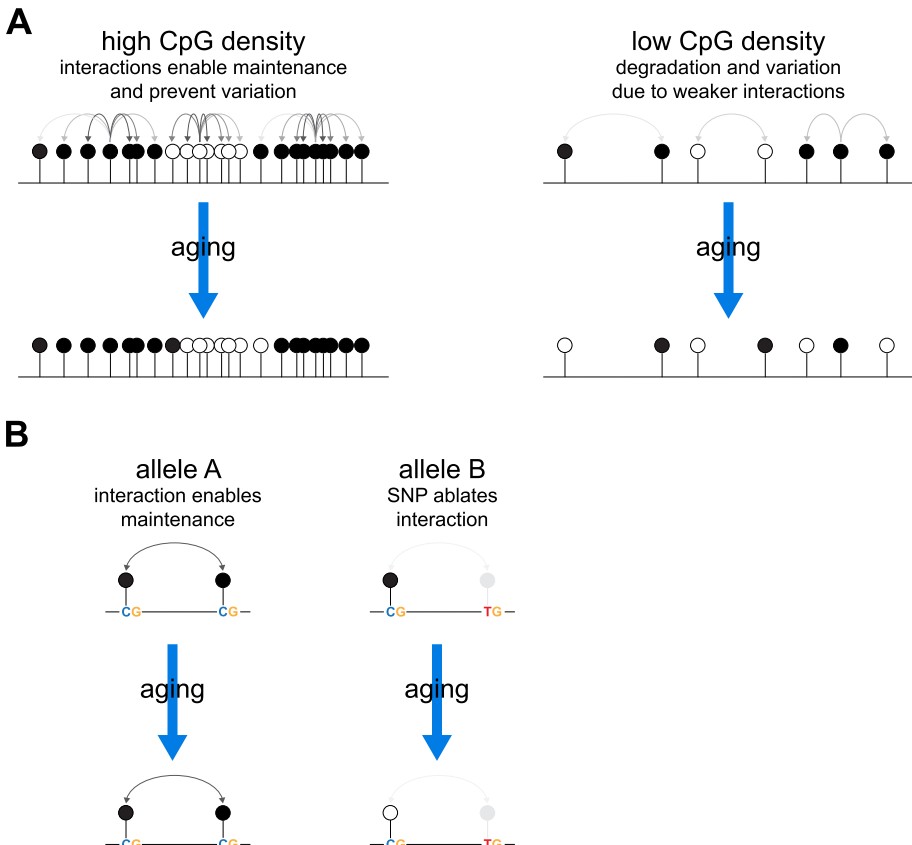

**Fig. 6** Local CpG density affects the trajectory of age-associated epigenetic changes. We propose that collaborative interactions between CpGs reinforce maintenance of methylation patterns in CpG dense regions (**a**). These interactions are weaker in CpG-poor regions leading to the degradation of methylation patterns with time and emergence of variation. These interactions can also be altered by SNPs leading to differences in epigenetic trajectories between individuals who inherit different alleles (**b**)

of change in DNA methylation with age. They suggested that this is instead primarily determined by the cellular environment. Here, we demonstrate that local DNA sequence does play a role in age-associated methylation changes by showing that local SNPs can alter methylation trajectories and that local CpG density affects how DNA methylation changes with age at individual CpGs in humans in vivo.

Our analyses in three groups of individuals show that lower CpG density regions display more variation in their trajectories with age than regions of high CpG density. Previous work has reported an overall increase in the disorder of DNA methylation patterns with age [23]. An analysis of entropy in Whole Genome Bisulfite Sequencing (WGBS) from a small number of samples also suggested that disorder increases with age at the single molecule level [53]. Specific loci which become more variable with age have also been defined in cross-sectional [23] and longitudinal [54] cohorts. These studies did not specifically examine CpG density of the loci reported. However, a longitudinal study of twins from birth to 18 months old reports that loci which vary between twins are located outside of CpG islands [55].

CpG density is known to be a determinant of steady-state DNA methylation patterns in cells. The most highly CpG dense portions of the genome are CpG islands which

are typically DNA methylation free [56]. Variation in CpG density alone can also predict the methylation levels of bacterial DNA fragments integrated into the genome of mouse embryonic stem cells [57]. Analysis of WGBS from human cell lines also revealed a strong relationship between CpG density and DNA methylation in heterochromatic PMDs independently of CpG islands [47]. These are typically gene and CpG poor compared to euchromatic portions of the genome [41]. They display disordered, variable patterns of methylation where at individual CpGs within PMDs, methylation levels are positively correlated with the surrounding CpG density [47]. PMDs overlap with the HiC defined nuclear heterochromatic B-compartment [42, 58]. These genomic regions have more disordered methylation patterns when quantified from single WGBS reads [53]. Analysis of diverse methylomes by WGBS also suggests that PMDs lose methylation with age [16]. However, these observations are all derived from static snapshots of individual samples from cell lines and tissues. Here we provide the first analysis of how CpG density affects DNA methylation changes with age observed longitudinally in humans in vivo.

Interactions and collaborative reinforcement of DNA methylation between adjacent CpG sites has been proposed as being vital to maintain DNA methylation patterns by mathematical modelling [59]. Detailed analysis of methylation dynamics in mES cells possessing a sole DNMT, DNMT1, also finds evidence of neighbor-guided error correction as being important in maintaining DNA methylation patterns [60]. These collaborative interactions between CpGs are likely to be strongly influenced by the distance between CpGs. Thus, in lower CpG density regions, the greater distance between CpGs could result in weaker collaboration between neighboring CpGs enabling greater degradation of developmentally established methylation patterns with age and the establishment of variation (Fig. 6a). These interactions are likely to be mediated by the molecular properties of the DNA methylation machinery. DNMT1 and DNMT3B both methylate processively along DNA strands whereas DNMT3A methylates in a distributive manner but can form multimers along the DNA fiber [61]. A computational analysis of DNA re-methylation dynamics suggests these processes are required to explain the observed rates of DNA re-methylation following replication [62]. The efficiency of DNMTs is also influenced by bases surrounding CpGs in vitro [63, 64] and in vivo [65]. These preferences could affect the efficiency by which some CpGs are methylated and could explain the effect of some of the SNPs we have uncovered that do not directly affect CpGs. As well as being the target of DNMTs, unmethylated and methylated CpG nucleotides are also bound by CXXC and Methyl-Binding Domain (MBD) respectively [66–69]. It is possible that these proteins also mediate the effects of CpG density on DNA methylation changes with age. CXXC proteins include TET1 which plays a role in demethylation as well as CFP1, MLL1, and MLL2 which deposit the histone modification H3K4me3 [70]. H3K4me3 inhibits the activity of de novo DNMTs through their ADD domains [71]. Thus, dense unmethylated CpGs can recruit proteins that reinforce their unmethylated status. It is unclear whether MBDs positively reinforce methylation patterns through binding to methylated CpGs; however, their binding in the genome tracks CpG density [72].

Genetic effects on steady-state DNA methylation levels have been widely documented in human populations as allele-specific methylation and meth-QTLs [24]. These are

hypothesized to reflect the alteration of TF binding by sequence polymorphisms with downstream effects on DNA methylation particularly at enhancers. TFs have been strongly implicated in programming DNA methylation in cell lines [27]. TF binding can be both promoted or hindered by DNA methylation [73]. However, the majority of TF binding sites in the genome have reduced methylation [74]. The hypothesis that changes in TF binding underpin meth-QTLs is also supported by a study showing that many *trans*-meth-QTLs correspond to TF genes [75] and the analysis of a subset of SNPs affecting TF binding motifs in lymphoblastoid cell lines [76]. We find that the majority of slope-QTLs are in intergenic, heterochromatic regions with a low CpG density. These regions are depleted in both genes and enhancers [41] suggesting slope-QTLs may be explained by alternative mechanisms. However, it is likely that slope-QTLs are not all explained by a single mechanism. 12.3% of slope-QTL CpGs are located in regions marked with enhancer-associated histone modifications in blood cells and these may be explained by TF-dependent mechanisms. However, we did not find strong evidence that any particular TF was strongly associated with slope-QTLs suggesting that if this is the case a range of TFs are involved.

Previous work has suggested that variation in the proportions of cell types present in the blood may underpin many changes in DNA methylation levels with age [22] and some epigenetic clocks are thought to capture similar changes [8]. However, another study has suggested that up to 151,537 loci tested display age-associated changes in DNA methylation when controlling for variation in cell type proportions [77]. Here, we find that between 9.77% and 64.60% of the 182,760 loci we find show age-associated changes in DNA methylation in LBC can be partly explained by variation in the proportions of cell types depending on whether we use directly measured or estimated cell counts. However, the proportion of variation explained by cell type variation at these CpGs is modest in both cases. While the majority of studies estimate cell type proportions from DNA methylation data, directly measured cell counts have the advantage of not been confounded by changes in cell-type-specific DNA methylation patterns with age (e.g. as recently described in monocytes) [78]. Furthermore, we observe that the effects of slope-QTLs and the association with CpG density we observe are not affected by correction for either directly measured or estimated blood cell counts.

In the current work, we have modelled methylation trajectories linearly. Previous studies of cultured fibroblasts [79] and cross-sectional human cohorts [80] suggest non-linear dynamics at some CpGs. Given the number of observations available per individual in the LBC cohort, non-linear trajectories cannot be fitted sufficiently robustly. By comparison to Generation Scotland, we find that the changes in methylation we observe in LBC are replicated in individuals aged over 65 but not in younger members of Generation Scotland. In younger individuals, we observe that low CpG density regions show an overall loss of DNA methylation but in older individuals low CpG density regions instead show variable trajectories between individuals. These observations are consistent with a non-linear trajectory for low CpG density regions over the life course. Previous work has shown an overall loss of DNA methylation with age [5]. This is particularly prominent at low CpG density intergenic regions [16] consistent with our observations in people under 65. The nature of the specific changes in DNA methylation that occur with age in older people remains understudied. One previous study of epigenetic clocks found that

their rate of change slows down in later life [81]. Taken together this suggests that the nature of DNA methylation changes at different regions varies non-linearly and changes at different points in the life course including in later life. The reasons for the difference in the behavior of low CpG density regions in older individuals remain unclear and warrant further investigation.

In addition to *cis*-slope-QTLs, we also find evidence for the existence of *trans*-slope-QTLs. Previous work has documented large numbers of *trans*-meth-QTLs [82] which have been ascribed to alterations in the expression of TFs or DNA methylation regulatory factors [75, 83]. It is likely that more *trans*-slope-QTLs exist. However, due to the large multiple testing burden incurred by searching every CpG against every possible SNP, it was not possible to detect large numbers of them in the current study. Future investigation of *trans*-slope-QTLs will require larger studies with a higher statistical power in combination with approaches which reduce the multiple testing burden by taking account of the lack of independence between SNPs [84].

## Conclusions

Taken together, our study suggests that DNA sequence, and CpG density in particular, has a major influence on the local tick rate of age-associated DNA methylation changes and how this varies between individuals. We ascribe this effect to interactions between neighboring CpGs reinforcing maintenance of methylation patterns through the action of the DNA methylation machinery.

## Materials and methods

### Statistical analysis

Statistical testing was performed using *R v4.0.2* unless otherwise stated. All tests were two-sided, unless otherwise stated. All linear models were fitted using the *lm* command from the base R package unless otherwise stated. Breusch-Pagan tests were conducted using the *bptest* function from the R package *lmtest* (*v4.02*). Further details of specific analyses provided in the relevant methods sections below.

### Cohort details

The Lothian Birth Cohort 1936 [31–33] is derived from a set of individuals born in 1936 who had mostly taken part in the Scottish Mental Survey 1947 at a mean age of 11 years as part of national testing of almost all children born in 1936 who attended Scottish schools on 4 June 1947. A total of 1091 participants who were living in the Lothian area of Scotland were re-contacted in later life. DNA methylation was measured for this cohort around 70 years of age and subsequently at a mean of 73, 76, and 79 years on Illumina 450 k arrays. In total, this corresponds to 2852 samples from 1056 unique individuals. In this study, we focused on the 600 individuals for whom ≥ 3 methylation measurements existed (283 females and 317 males). A breakdown of the sample demographics can be found in Table 1.

The Generation Scotland dataset was derived from a subset of individuals in the Generation Scotland or Scottish Family Health Study cohort. Generation Scotland is a family-based population cohort investigating the genetics of health and disease in approximately 24,000 individuals across Scotland [37, 85]. Baseline data were collected

between 2006 and 2011. Two sets of individuals are analyzed here. These two sets contain 5101 and 4450 individuals respectively for whom Illumina EPIC array data had been collected from blood at baseline contact [37]. The age ranges of participants in the two sets were 18–95 and 18–93 respectively.

All participants provided written informed consent. Ethical permission for the Lothian Birth Cohort 1936 study protocol was obtained from the Multi-Centre Research Ethics Committee for Scotland (Wave 1: MREC/01/0/56), the Lothian Research Ethics Committee (Wave 1: LREC/2003/2/29), and the Scotland A Research Ethics Committee (Waves 2–4: 07/MRE00/58). All components of Generation Scotland received ethical approval from the NHS Tayside Committee on Medical Research Ethics (REC Reference Number: 05/S1401/89). GS has also been granted Research Tissue Bank status by the East of Scotland Research Ethics Service (REC Reference Number: 20/ES/0021), providing generic ethical approval for a wide range of uses within medical research.

### Processing of Illumina Infinium array data

Infinium arrays from both the LBC and Generation Scotland cohorts were processed from IDAT files. These were normalized using the *ssNoob* method from the Bioconductor package *minfi* (*v1.22.1*) to derive beta values and detection *p*-values (beta threshold = 0.001) [86, 87]. Individual beta values were excluded where detection *p*-value was > 0.01. Infinium probe locations in the hg38 genome build were taken from Zhou et al. [34]. Probes categorized as overlapping common SNPs or having ambiguous genome mapping in that paper were excluded from the analysis (*MASK.snp5.common, MASK.mapping, MASK.sub30.copy* from Zhou et al.) [34]. Non-CG probes and probes not located on autosomes were also excluded from the analysis.

### Processing of SNP data

DNA samples from the Lothian Birth Cohort 1936 were genotyped at the Wellcome Trust Clinical Research Facility using the Illumina 610-Quadv1 array (San Diego) [88]. Individuals were excluded based on relatedness ($n=8$), unresolved sex discrepancy ($n=12$), low call rate ($\leq 0.95$ $n=16$), and evidence of non-European descent ($n=1$). SNPs were included if they had a call rate $\geq 0.98$, a minor allele frequency $\geq 0.01$, and a Hardy–Weinberg equilibrium test with $p \geq 0.001$.

### Modelling of DNA methylation trajectories

DNA methylation trajectories for each CpG in each individual were modelled by fitting linear models of beta value with age for each CpG and individual using R:

$$beta_{ij} = \alpha_{ij} age_i + \gamma_{ij} \tag{1}$$

where $beta_{ij}$ is the beta value of CpG $j$ for individual $i$, $age_i$ is the age of individual $i$, $\alpha_{ij}$ is the age effect for CpG $j$ in individual $i$, and $\gamma_i$ is the intercept for CpG $j$ in individual $i$. This was only done for individuals and CpGs for which $\geq 3$ datapoints were present in the processed dataset. Slopes for each individual and CpG were taken from the linear

models ($\alpha_{ij}$). Mean slopes were calculated as the mean of $\alpha_{ij}$ across all $N$ individuals for CpG $j$:

$$\mu_j = \frac{1}{N} \sum_{i=1}^{N} \alpha_{ij} \tag{2}$$

CpGs were considered to have a rate of change significantly different from 0 if the distribution of slopes for all individuals had a Bonferroni-corrected $p$-value < 0.01 by $T$-test. We defined rapid gain CpGs as those with significant slopes where the mean change in Beta per year was greater than the local minimum on a histogram of all CpGs with a significant slope (bin size = 0.0005 and threshold > 0.0159, both in beta/year).

When modelling methylation trajectories in the Generation Scotland dataset, linear models of beta with age were calculated across all individuals present in the analysis and slope coefficients and residuals extracted. To compare individuals of a similar age to those in LBC we used only the 406 or 519 individuals aged > 65 years from Generation Scotland set 1 and 2 where indicated.

To correct for variation in blood cell populations, we used white blood cell counts (neutrophils, lymphocytes, monocytes, eosinophils, and basophils) for each sample collected on a Sysmex HST system under standard operating procedures within the National Health Service Haemotology laboratory, Western General Hospital, Edinburgh. We then used these to derive residualised beta values corrected for variation in these blood cell populations by fitting a linear model:

$$beta \sim n_C + l_C + m_C + e_C + b_C \tag{3}$$

where: $n_c$ = neutrophil count, $l_c$ = lymphocyte count, $m_c$ = monocyte count, $e_c$ = eosinophil count, and $b_c$ = basophil count. The residuals of this model were then used as corrected beta values and methylation trajectories were then modelled from them as above.

Blood cell type counts were also estimated for the LBC samples from DNA methylation levels using the Houseman approach [39] as previously described [89]. This approach resulted in estimated counts for granulocytes, B-cells, CD4 T-cells, CD8 T-cells, natural killer cells, and monocytes. Beta values were then corrected for these counts by deriving residualised beta values in a manner similar to that used for the directly measured cell counts.

The variability in methylation trajectories at each CpG was investigated by calculating the variance of all individual linear model slope coefficients for that CpG. As the variance was strongly related to the mean beta value of each CpG (Additional file 1: Fig S4f), we normalized variance of CpGs by calculating the median of 20 equal bins based upon the mean beta and then subtracting the calculated median from all the CpGs in that bin. These residualised variances were then analyzed.

**Mixed effect models of DNA methylation trajectories**

The lme4 package (v1.1–28) in R (v4.1.2) was used to model the beta value as a linear mixed effects model with random intercepts, where age was incorporated as a fixed effect and samples were group based on the individual, leading to the equation:

$$beta_{ij} = \alpha_j age_i + \gamma_i + \varepsilon_{ij} \qquad (4)$$

where $beta_{ij}$ is the beta value of CpG $j$ for individual $i$, $age_i$ is the age of individual $i$, $\alpha_j$ is the age effect for CpG $j$, $\gamma_i$ is the random intercept for individual $i$, and $\varepsilon_{ij}$ is the random error associated with CpG $j$ for individual $i$. Data corresponding to the 600 individuals from the Lothian Birth Cohort for whom three or more methylation measurements existed was used for this model. Of the 345,895 reliably measured autosomal CpGs for which analysis was attempted, 345,890 had sufficient data to fit a mixed model.

**Analysis of variation in cell type proportions on age-associated DNA methylation changes**

Analysis was conducted using the *lm* base package in R (v4.1.2) to investigate the extent to which the change in beta value with age can be explained by changes in the proportion of cell types with age. The rate of change with age in each cell-type proportion was estimated for the 587 individuals for whom at least three methylation and at least three direct cell-type count measurements (for five different cell types) were available. For each individual and time point, the cell counts were used to calculate the proportion of each cell type contained in the sample. The proportion of each cell type was fit as a linear model of age:

$$p_{ij} = \Delta p_{ij} * age_i + \varepsilon_{ij}, \qquad (5)$$

where $p_{ij}$ is the proportion of cell type $j$ for individual $i$, $\Delta p_{ij}$ is the rate of change of cell type $j$ for individual $i$, $age_i$ is the age of individual $i$, and $\varepsilon_{ij}$ is the intercept associated with cell type $j$ for individual $i$.

For each of the 182,760 CpGs that show significant change in methylation with age, the rate of change in beta was fit as a linear model of the rates of change associated with the different cell types:

$$\Delta \beta_{ik} = \sum_{j=1}^{4} c_{jk} \Delta p_{ij} + \epsilon_k, \qquad (6)$$

where $\Delta \beta_{ik}$ is the rate of change of CpG $k$ for individual $i$, $c_{jk}$ is the effect of the rate of change in cell type $j$ for CpG $k$, and $\epsilon_k$ is the intercept associated with CpG $k$. Only four of the five cell types were included in this model since the proportions of all cell types sum to one. Since the proportion of lymphocytes was highly correlated with the proportion of neutrophils, lymphocytes were chosen to be removed from the analysis. Variance inflation factors indicated that the remaining variables were not highly correlated with each other. For each CpG, the $p$-value (Bonferroni-corrected, $p < 0.01$) and $R^2$ value were extracted from the model to evaluate the ability to explain changes in methylation by the changes in cell types.

The analysis above was then repeated, with the cell-type proportions derived from measured cell counts being replaced by cell-type proportions estimated by the Houseman algorithm.

### Analysis of epigenetic clocks

Horvath epigenetic clock CpGs were taken from Horvath 2013 Supplementary Table 3 [6]. This table contains details of the age relationship these CpGs displayed in the original derivation of this clock [6]. We defined Horvath clock increasing and decreasing CpGs from these coefficients. Hannnum epigenetic clock CpGs were taken from Hannum et al. Supplementary Table 3 [7]. As this did not include details of the rate of change at each clock CpG over time, we determined increasing and decreasing CpGs using the Generation Scotland set 1 cross-sectional dataset by fitting linear models to each as described above.

### Analysis of CpG annotation

CpGs were annotated to CGIs based on Illingworth et al. [90]. Overlapping CGI intervals were merged using BEDtools (*v2.27.1*) [91] before they were converted to hg38 positions using the UCSC browser *liftover* tool. CpGs were then overlapped with CGIs using BEDtools. CGI shores were defined as the 2 kb on either side of each CGI using BEDtools and similarly overlapped with CpGs. CpGs were annotated relative to genes using BEDTools to overlap them with ENSEMBL protein coding genes (Ensembl Release 98/GCRh38). CpGs were annotated as being located at a transcription start site (TSS) if they overlapped a protein coding TSS and as located in a gene body if they overlapped a transcript but not a TSS. The remaining CpGs which did not overlap a TSS or transcript were annotated as intergenic. PMD and highly methylated region (HMD) definitions were taken from Zhou et al. [16] from *commonPMDs* and *commonHMDs* defined across 40 tumor and 9 normal samples and overlapped with CpGs using BEDtools.

Infinium probes were mapped to existing ChromHMM annotations [92] using the BEDtools *intersect* function (*v2.27.1*) [91]. Identical ChromHMM labels were merged for analysis. To test for enrichment of an annotation, Fisher's exact test was performed for number of rapid gain CpGs or slope-QTL CpGs against number of control probes. ENCODE GM12878 ChromHMM [43] annotations were downloaded as bed-files from the UCSC genome browser. Roadmap Epigenomics ChromHMM annotations for primary human cell types were downloaded as mnemonics BED files from the Roadmap Epigenomics site [44, 93]. The 23 primary blood cell types analyzed here were selected by manual examination of Roadmap Epigenomics sample annotations.

To calculate local CpG density, windows of sequence (e.g., $-/+300$ bp) were extracted around each CpG analyzed from the hg38 genome sequence using BEDtools (*v2.23.0*) and the number of CpG dyads within this window counted.

### Slope-QTL analysis

Associations between genotype and local rates of methylation change (quantified as the slope coefficient for each individual linear model) were analyzed using the *matrix-eQTL R* package (*v2.23*) [94]. For *cis*-associations, we set a distance cut-off of 1 Mb. Significant associations were those where the Benjamini-Hochberg-corrected *p*-value was < 0.05. The analysis of *trans*-associations was carried out similarly with the distance threshold removed. We removed loci where apparent differences in the rate of change associated with genotype were caused by effects on the variability of the slope because this could result from technical effects of SNPs on the hybridisation of Infinium probes. To do so, we tested for associations between SNPs within 1 Mb and the variance of methylation at each CpG (quantified as the residual sum of each individual linear model). CpG-SNP pairs with a significant association to variability were defined as those with a matrix-eQTL Benjamini-Hochberg-corrected $p < 0.05$.

We then performed a conditional analysis to determine how many independent SNP-CpG pairs were present. The SNP with the most significant *p*-value associated with each CpG was designated the lead SNP, and then all other associated SNPs were tested against the residuals of the linear model of the lead SNP. Other SNPs were then designated as independent hits if their *p*-value of association was < 0.05 following Bonferroni correction.

### Age × genotype interaction modelling

A linear model (methylation ~ age × genotype) was fitted to each CpG in LBC. The correlation between these age × genotype effect sizes and the rate of methylation change ~ genotype effect size in the LBC was calculated using Pearson's coefficient.

### Overlap of slope-QTLs with GWAS catalog

Slope-QTL SNPs were overlapped with the NHGRI-EBI GWAS catalog downloaded as a text file on 7 January, 2022 [46] The following traits were defined as associated with blood traits: basophil count, eosinophil counts, eosinophil percentage of white cells, erythrocyte sedimentation rate, hematocrit, IgE levels, lymphocyte counts, lymphocyte percentage of white cells, mean corpuscular hemoglobin, mean corpuscular volume, monocyte count, neutrophil percentage of white cells, neutrophil-to-lymphocyte ratio, platelet count, platelet distribution width, plateletcrit, white blood cell count, white blood cell count (monocyte).

### Analysis of TF binding sites at slope-QTLs

ENCODE-defined TF binding clusters were downloaded from the UCSC browser [95] and filtered for those present in GM12878 cells using the unix *grep* command before being overlapped with CpG and SNP probes using the BEDtools *intersect* function (*v2.27.1*) [91]. Counts of individual TF binding sites overlapping slope-QTL CpGs and SNPs were then compared to the background of all probes in their respective analyses using Fisher's exact tests. Binding sites for the following ENCODE TFs were removed from the analysis because they are chromatin modulating factors rather than

sequence-specific TFs: POLR2A, ARID3A, RAD21, EZH2, SUZ12, HDAC6, SIN3A, HDAC2, BMI1, CBX3, KDM1A, ASH2L, EP300, SMC3, CBX5, CBX3, KAT2A.

## Supplementary Information

---

Additional file 1: Supplementary figures Fig. S1-5 and legends.

Additional file 2: Supplementary tables Table S1-7.

Additional file 3. Review history.

---

### Acknowledgements

We thank Riccardo Marioni, Chris Haley, Ailith Ewing, David Porteous, Chris Ponting, Rob Illingworth, Tamir Chandra, Sara Hagg, Yunzhang Wang, Chantriolnt-Andreas Kapourani, Nick Gilbert, Hannes Becher and members of the Sproul lab for helpful discussions about the study and the manuscript. This work has made use of the resources provided by the University of Edinburgh digital research services and the MRC IGC compute cluster. We are grateful to all the families who took part in the Generation Scotland study along with the general practitioners and the Scottish School of Primary Care for their help in recruiting them, and the entire Generation Scotland team, which includes interviewers, computer and laboratory technicians, clerical workers, research scientists, volunteers, managers, receptionists, healthcare assistants, and nurses.

### Peer review information

Anahita Bishop and Kevin Pang were the primary editors of this article and managed its editorial process and peer review in collaboration with the rest of the editorial team.

### Review history

The review history is available as Additional file 3.

### Authors' contributions

JH, LK, and DS conducted the computational analysis presented in the manuscript. RMW, SEH, SRC, DMH, and ELH curated data relating to the study. QC and PMV contributed to the analysis and interpretation of results. ALS, JDS, GDW, ADM, KLM, AMM, and IJD ascertained subjects and obtained samples and funding for the profiling of cohort samples. DS planned and supervised the study and wrote the manuscript with input from JH and review by all authors. All author(s) read and approved the final manuscript.

### Funding

DS is a Cancer Research UK Career Development fellow (reference C47648/A20837), and work in his laboratory is also supported by an MRC university grant to the MRC Human Genetics Unit. LK is a cross-disciplinary postdoctoral fellow supported by funding from the University of Edinburgh and Medical Research Council (MC_UU_00009/2). S.R.C. and I.J.D. were supported by a National Institutes of Health (NIH) research grant R01AG054628, and S.R.C is supported by a Sir Henry Dale Fellowship jointly funded by the Wellcome Trust and the Royal Society (221890/Z/20/Z). AMM is supported by the Wellcome Trust (104036/Z/14/Z, 216767/Z/19/Z, 220857/Z/20/Z) and UKRI MRC (MC_PC_17209, MR/S035818/1). PMV acknowledges support from the Australian National Health and Medical Research Council (1113400) and the Australian Research Council (FL180100072). DMH is supported by a Sir Henry Wellcome Postdoctoral Fellowship (Reference 213674/Z/18/Z). We thank the LBC1936 participants and team members who contributed to the study. Further study information can be found at https://www.ed.ac.uk/lothian-birth-cohorts. The LBC1936 is supported by a jointly funded grant from the BBSRC and ESRC (BB/W008793/1), and also by Age UK (Disconnected Mind project), the Medical Research Council (G0701120, G1001245, MR/M013111/1, MR/R024065/1), and the University of Edinburgh. Genotyping of LBC1936 was funded by the BBSRC (BB/F019394/1), and methylation typing of LBC1936 was supported by Centre for Cognitive Ageing and Cognitive Epidemiology (Pilot Fund award), Age UK, The Wellcome Trust Institutional Strategic Support Fund, The University of Edinburgh, and The University of Queensland. Work on Generation Scotland was supported by a Wellcome Strategic Award "STratifying Resilience and Depression Longitudinally" (STRADL; 104036/Z/14/Z) to AMM, KLE, and others, and an MRC Mental Health Data Pathfinder Grant (MC_PC_17209) to AMM. Generation Scotland received core support from the Chief Scientist Office of the Scottish Government Health Directorates (CZD/16/6) and the Scottish Funding Council (HR03006). DNA methylation profiling and analysis of the GS:SFHS samples was supported by Wellcome Investigator Award 220857/Z/20/Z and Grant 104036/Z/14/Z (PI: AM McIntosh) and through funding from NARSAD (Ref: 27404; awardee: Dr DM Howard) and the Royal College of Physicians of Edinburgh (Sim Fellowship; Awardee: Dr HC Whalley).

### Availability of data and materials

According to the terms of consent for Lothian Birth Cohort 1936, data are available on request from the Lothian Birth Cohort Study, University of Edinburgh (https://www.ed.ac.uk/lothian-birth-cohorts/data-access-collaboration). Similarly, according to the terms of consent for Generation Scotland participants, access to data must be reviewed by the Generation Scotland Access Committee. Applications should be made to access@generationscotland.org.

## Declarations

### Ethics approval and consent to participate
All participants provided written informed consent. Ethical permission for the Lothian Birth Cohort 1936 study protocol was obtained from the Multi-Centre Research Ethics Committee for Scotland (Wave 1: MREC/01/0/56), the Lothian Research Ethics Committee (Wave 1: LREC/2003/2/29), and the Scotland A Research Ethics Committee (Waves 2–4: 07/MRE00/58). All components of Generation Scotland received ethical approval from the NHS Tayside Committee on Medical Research Ethics (REC Reference Number: 05/S1401/89). GS has also been granted Research Tissue Bank status by the East of Scotland Research Ethics Service (REC Reference Number: 20/ES/0021), providing generic ethical approval for a wide range of uses within medical research.

### Competing interests
AMM has received speaker fees from Illumina and Janssen, and research funding support from The Sackler Trust. JDS has received funding via an honorarium associated with a lecture for Wyeth and funding from Indivior for a study on opioid dependency.

### Author details
[1]MRC Human Genetics Unit, Institute of Genetics and Cancer, University of Edinburgh, Edinburgh, UK. [2]Institute for Molecular Bioscience, University of Queensland, Brisbane, QLD, Australia. [3]Present address: Wellcome Sanger Institute, Hinxton, Cambridgeshire, UK. [4]Centre for Genomic and Experimental Medicine, Institute of Genetics and Cancer, University of Edinburgh, Edinburgh, UK. [5]Present address: School of Psychology, University of Exeter, Edinburgh, UK. [6]Department of Psychology, Lothian Birth Cohorts Group, University of Edinburgh, Edinburgh, UK. [7]Social, Genetic and Developmental Psychiatry Centre, Institute of Psychiatry, Psychology and Neuroscience, King's College London, London, UK. [8]Division of Psychiatry, University of Edinburgh, Royal Edinburgh Hospital, Edinburgh, UK. [9]Aberdeen Biomedical Imaging Centre, Institute of Medical Sciences, University of Aberdeen, Aberdeen, UK. [10]Division of Imaging Science and Technology, Medical School, University of Dundee, Dundee, UK. [11]CRUK Edinburgh Centre, Institute of Genetics and Cancer, University of Edinburgh, Edinburgh, UK.

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

## 
