## [Additional file 3. Review history. · Genome Biology]

Review History

First round of review

Reviewer 1

Are you able to assess all statistics in the manuscript, including the appropriateness of statistical tests used? No, I do not feel adequately qualified to assess the statistics.

Comments to author:

In this manuscript, the authors have performed longitudinal analysis of the epigenetic clock in a large human cohort. Specifically, they find some CpGs at which the rate of change accelerates in later life. They then characterise these CpG sites.

Overall, with the proviso that I am not really qualified to assess the statistics, I think this is a powerful study. The longitudinal analyses in >600 individuals is definitely of note. I only have a few comments listed below:

1. I would like to see more analysis of the relevant SNPs themselves. After all, the core message of the paper is about how sequence affects methylation ageing trajectories. Have these SNPs been implicated in diseases or phenotypes? Do they alter TF binding sites? I think a section exploring this would add great value to the ms. Just requires some more analysis, no more data generation.

2. I don't understand the last sentence of the Results - "In addition, our results also suggest that alterations in a CpG's local sequence context caused by SNPs can alter its methylation trajectory with age". Are they saying that it is not the SNPs that are relevant but other sequence that they change?

3. Can the authors use their data to estimate roughly what proportion of the known epigenetic clock is influenced by SNPs? This will also be relevant for Reference 25 in that, how much is trans environment and how much is sequence? For example, it is known that lifespan altering interventions can alter the epigenetic ageing clock i.e. 'trans' factors.

4. Can they use their longitudinal data to determine if there are unlinked CpG sites that show coordinated ageing methylation dynamics? That is, if CpG X goes up in any given individual, then unlinked CpG Y will also go up. If CpG X doesn't change then neither does CpG Y.

Reviewer 2

Are you able to assess all statistics in the manuscript, including the appropriateness of statistical tests used? Yes, and I have assessed the statistics in my report.

Comments to author:

The article describes results from a longitudinal DNA methylation analysis based on 600 individuals aged 67-80 yrs. The authors reported a subset of loci that rapidly changed during this age interval. A small fraction of such loci showed genotype-dependent epigenetic aging effects, and such loci were predominantly mapping to the regions of low DNA methylation density. The authors are asking

interesting questions but the study contains significant limitations.

There are numerous statements which are not necessarily incorrect but do lack precision. Several examples from the Abstract only:

1. "DNA methylation is an epigenetic mark associated with gene repression and genome stability." Does association between DNA meth and gene repression apply to gene bodies? Regarding the stability, genomes of Drosophila and yeast (which lack DNA methylation) - are they less stable compared to the other genomes which are made of modified DNA?

2. "Its pattern in the genome is disrupted with age and these changes can be used to statistically predict age with epigenetic clocks." Why is the term "disrupted" used here? They just change with age, and we don't know yet if such changes have only negative impact. Some of them can be adaptive, and therefore useful.

3. "Rates of aging inferred from these clocks correlate with human health." What is correlation with "health"? If some associations between accelerated age and disease are meant, it should be stated appropriately.

4. "However, the molecular mechanisms underpinning age associated DNA methylation changes are unknown." It suggests that mechanisms will be investigated but the study is totally descriptive.

5. "Local DNA sequence plays a strong role in programming DNA methylation levels at individual loci independently of age". It makes an impression that the age-dependent impact of DNA sequence [variation] on DNA methylation levels has been investigated but, in fact, this question is quite new. The paper would benefit significantly if the key question is better explained.

Page 4.

The decision to limit the study to individuals with 3 datapoints, discarding more than 400 samples, seems to be a counterproductive given that the key question was how DNA sequence variation contributes to aging trajectories of DNA methylation. The larger the sample, the higher the power, the more robust results.

Lines 28-47. Also not clear why the detected changes were compared to the cytosines from the epigenetic clock. What does it say about the reliability of measurements? The Horvath clock is generated of maximally informative cytosines which do not necessarily strongly correlate with age.

Page 5.

1. Lines 21-25. Correction for blood cell count differences - was it done just in this one instance as a "check" and the resulting figure is added to the supplement (Sup Fig 2 b). ? If it was not done before any downstream analysis, age related cellular effects can be a major confounder. This could be the main reason for inconsistencies with previous epigenetic aging studies, including the predominant age-dependent gain of DNA methylation marks (lines 9-10). This is a major problem with this study which requires a full re-analysis.

2. Continuing with the theme of cell count correction - according to the methods all the lymphocytes were grouped under one cell type. The most popular cell-count correction method for this type of data (Illumina 450K arrays) - the Houseman's algorithm with Reinus et.al. dataset as a reference - includes 4 separate lymphocyte fractions - CD4 T-cells, CD8 T-cells, B-cells and Natural Killers - all accounted for independently. Again, what is the reason for deviating from such standard practice and grouping all these

cell types together?

3. Illumina 450K DNA methylation arrays are known to contain batch effects, in particular - sentrix ID and sentrix row position have been shown to have an influence on the estimated DNA modification levels. The paper, however, doesn't say anything about these. It is necessary to know if the sentrix IDs were randomised between the first measurement, and the follow up measurements within the longitudinal design. If the first batch of measurements were placed on one set of arrays, and later follow up experiments were interrogated using another set of arrays, then there is a high chance that some of the results are caused by technical variance.

4. One general question: what was the point of focusing on the rapid gain and ignoring weaker but still significant aging effects? This hardly understandable choice takes out the context of previous aging studies, making it hard to compare current results with what has been known in the field.

5. Furthermore, low DNA modification regions that start aging at 67-80 years of age, is quite unusual. It would be necessary to show if this finding can be replicated in a study with donors collected from a broader age interval. In particular - how the aging profiles of the obtained set of CpGs exhibiting rapid DNA modification changes look like in other public datasets.

6. Line 32. "Large hypomethylated genomic regions termed partially methylated domains..."
Hypomethylated compared to what? Are they "hypomethylated" or PMDs?

Reviewer 3

Are you able to assess all statistics in the manuscript, including the appropriateness of statistical tests used? Yes, and I have assessed the statistics in my report.

Comments to author:

In this manuscript, Higham et al. investigate age-related changes in CpG methylation genome-wide, and the genetic basis of such changes, using data from 600 individuals from the Lothian Birth cohort. The main findings are (1) a set of CpGs where there are slope-QTLs; ie. a genetic variation influences the extent to which methylation at a nearby CpG changes with age. (2) that CpGs in regions of low CpG density are particularly prone to change with age.

Currently, we know that DNA methylation can be used to predict chronological age, but we have little-to-no understanding of why? This paper advances our knowledge of this.

My major concern is technical in nature; the authors have not convinced me that their set of CpGs claimed to be associated with age, are indeed associated with age. I am concerned about cell type composition, as well as replication / can these CpGs actually predict age? The value of the entire paper rests on providing strong, positive answers to these questions. I will expand on these critical questions in the next two paragraphs.

We know that cell type composition change with age. This has led to a criticism of the Hannum clock (Jaffe and Irizarry, Genome Biology, which should really be discussed in the author's manuscript). First, I note that the authors find that ~50% of CpGs they examine, are also

significantly associated with age. That seems to be an astronomical number. While it could be true, the magnitude of this number really suggests confounding by cell type. In response to this, the authors have examined if the age effect disappears when you control for white blood cell counts (an imperfect measure of cell type composition). They claim all their CpGs are still significant when you control for WBC. However, when I compare Fig 2a to Supplemental Fig 2b I see a 10x reduction in effect size (beta value). It is really hard for me to interpret this in any other way than the present study is severely affected by cell type composition. Strangely, in my understanding, the QTL analysis does not correct for WBC which should be a minimum (but perhaps I have misunderstood the manuscript).

Related, I was looking for evidence that the ~185k age-associated CpGs can be replicated or used to predict age or something else will support the notion that these are actually "true" age-associated CpGs. The reason we (the field) really like Horvath's clock is that it has been consistently replicated. There is some minimal effort spent on this in the manuscript, but the effort falls far short of what I think is required. You could look at replication in a different cohort. You could build an age clock using random subsets of these CpGs and validate this random clock in a new cohort (and repeat for many random samples). However, even if it replicates, it could still be cell type composition.

Together, these two issues suggests to me that the work is technically flawed. It is not clear to me what the right approach is to satisfy my concerns about these issues; I hope the authors can do so. Having said that, the idea of the paper is quite interesting and worth pursuing.

Aside from these (potential) technical flaws, here is the usual list of issues

- * It would be natural to use the Horvath / Hannum clock to predict age in the data in the paper and assess the predictors. I am especially interested in the Hannum predictor which is less widely used. It would be interesting to look at the epigenetic age "acceleration" (ie. the residual) and see if it is consistent or changes between samples from the same individual.

- * The authors should explicitly provide the regression model of beta value on age that they fit to estimate the mean changes of methylation per year.

- * I would encourage the authors to consider mixed effects models to account for the correlation between samples from the same individual. At the minimum, I would like to see some discussion of why they took their approach and not one of many alternatives.

- It could be very interesting to see if: (a) different individuals differ with regards to their epigenetic age trajectories at the measured time points, similar to the way different individuals can have different trajectories for individual CpGs (b) there are genetic variants associated with such differences in epigenetic age trajectory

- * With respect to the individual CpGs: (a) are there any known TF motifs overlapping CpGs that show age-associated changes? If so, is the expression of these TFs associated (even if the association is weak) with age (e.g. in GTEx)? This is relevant since abnormal methylation can compromise (or sometimes increase) TF binding, which would essentially mimic reduced (or increased) expression level of the TF.

- (b) It would be very informative to do the QTL analysis but focusing only on variants within the coding, and nearby regulatory, regions of epigenetic machinery (especially DNA methylation machinery) genes. This becomes particularly relevant given the (plausible) model of disrupted maintenance at CpG-poor regions that the authors put forward at the end

* As the authors mention in the intro, many changes of methylation occur at CpGs present in CGIs bound by PRC2. Therefore, for figures 2d and 4a, they should separately treat CGIs (and CGI shores) for housekeeping genes and genes which are targets of PRC2 (and often expressed in a tissue/developmental-stage-specific manner).

I have not assess data availability as per Genome Biology standards.

Response to reviewers GBIO-D-21-01234 Higham *et al*

We thank the reviewers for recognizing the importance of our work and for their supportive and insightful comments that have helped us substantially improve the manuscript.

We enclose a revised copy of the manuscript with changes highlighted in **blue**.

In addition to your comments, we received editorial advice in the decision letter (comments are highlighted in *blue italics* throughout this text):

Thank you very much for submitting your manuscript entitled 'Local CpG density affects the trajectory of age-associated epigenetic changes' to Genome Biology, and please accept my apologies for the delay in replying to you about it. It has now been seen by three referees and their comments are accessible below.

As you will see from the reports, the referees find the manuscript of potential interest, but they raise serious concerns that the study needs more support of the CpGs associated with aging, by validation and clarifications on cell type confounders, as noted specifically by Referees 2 and 3. It seems to us to be essential that all of the referees' concerns are fully addressed, in the form of a revised manuscript, before we can reach a final decision on publication.

In revising the study we have addressed all the comments and especially focused on understanding the implications of our findings across a broader range of ages and clarifying the relationship between the differences in DNA methylation levels we observe and variation in the proportion of different cell types in blood in accordance with the reviewer's suggestions.

Reviewer 1:

In this manuscript, the authors have performed longitudinal analysis of the epigenetic clock in a large human cohort. Specifically, they find some CpGs at which the rate of change accelerates in later life. They then characterise these CpG sites.

Overall, with the proviso that I am not really qualified to assess the statistics, I think this is a powerful study. The longitudinal analyses in >600 individuals is definitely of note. I only have a few comments listed below:

We thank the reviewer for their kind comments highlighting the importance of the study.

1. I would like to see more analysis of the relevant SNPs themselves. After all, the core message of the paper is about ow sequence affects methylation ageing trajectories. Have these SNPs been implicated in diseases or phenotypes? Do they alter TF binding sites? I think a section exploring this would add great value to the ms. Just requires some more analysis, no more data generation.

As the reviewer suggests we have included further analysis of the lead SNPs associated with slope-QTL SNPs.

To ask whether lead slope-QTL SNPs had previously been associated with human traits, we cross-referenced them with the NHGRI-EBI GWAS Catalog catalogue and report our analysis in the manuscript as follows:

We found that 93 (6.66%) of the slope-QTL SNPs were present in the NHGRI-EBI GWAS catalogue (Supplementary Table 4) (46). The most frequent annotation was for SNPs associated with 'Hip

circumference adjusted for BMI' (8 SNPs). In addition, 20 of the SNPs were annotated as being previously associated with blood cell traits such as 'Platelet count' and 'Eosinophil counts' (for full list see methods).

We also asked the degree to which slope-QTL lead SNPs might affect TF binding sites by comparing them with TF binding sites defined for ENCODE in GM12878 cells as follows:

We similarly analysed the lead SNPs associated with slope-QTLs to ask if they overlapped TF binding sites in GM12878. In this analysis, the binding sites for 14 TFs were significantly enriched at the SNPs (Supplementary Table 7). However, even the most significantly enriched, CTCF, was only observed at 4.44% of the slope-QTL SNPs. Overall, this suggests that unlike cis-meth QTLs, the majority of cis-slope-QTLs are not associated with TF binding sites.

2. I dont understand the last sentence of the Results - "In addition, our results also suggest that alterations in a CpGs local sequence context caused by SNPs can alter its methylation trajectory with age". Are they saying that it is not the SNPs that are relevant but other sequence that they change?

We have clarified this sentence to:

This suggests that slope-QTLs are found within low CpG density regions and SNPs associated with them frequently alter the regions CpG density or the bases neighbouring CpGs.

3. Can the authors use their data to estimate roughly what proportion of the known epigenetic clock is influenced by SNPs? This will also be relevant for Reference 25 in that, how much is trans environment and how much is sequence? For example, it is known that lifespan altering interventions can alter the epigenetic ageing clock ie 'trans' factors.

Epigenetic clock CpGs are those which show consistent behaviour across individuals. That is, they have a consistent relationship between the beta value at that probe and chronological age. The slope QTLs we have uncovered show an inconsistent relationship between the probe's beta value and age, one that is dependent on the genotype at a nearby SNP. We would therefore expect slope QTL CpGs to be depleted in loci that are part of epigenetic clocks.

To ask if this was the case, we overlapped slope QTL CpGs with epigenetic clock CpGs. We find that only 1 CpG from the combined list of 399 Hannum and Horvath clock CpGs is found in the 1,456 CpGs found in slope QTLs.

4. Can they use their longitudinal data to determine if there are unlinked CpG sties that show co-ordinated ageing methylation dynamics? That is, if CpG X goes up in any given individual, then unlinked CpG Y will also go up. If CpG X doesnt change then neither does CpG Y.

In principle we could use the longitudinal data to perform such an analysis. However, to do so robustly in a manner that is independent of batch effects would necessitate the development of an entirely new methodology and so is beyond the scope of the period granted for revisions.

Reviewer 2:

The article describes results from a longitudinal DNA methylation analysis based on 600 individuals aged 67-80 yrs. The authors reported a subset of loci that rapidly changed during this age interval. A small fraction of such loci showed genotype-dependant epigenetic aging effects, and such loci were predominantly mapping to the regions of low DNA methylation density. The authors are asking interesting questions but the study contains significant limitations.

We thank the reviewer for their useful comments. We hope that our revisions have clarified the manuscript and provided additional evidence supporting our claims.

There are numerous statements which are not necessarily incorrect but do lack precision. Several examples from the Abstract only:

We have edited the abstract to clarify these statements as detailed below.

1. "DNA methylation is an epigenetic mark associated with gene repression and genome stability." Does association between DNA meth and gene repression apply to gene bodies? Regarding the stability, genomes of Drosophila and yeast (which lack DNA methylation) - are they less stable compared to the other genomes which are made of modified DNA?

We have clarified this sentence as follows:

DNA methylation is an epigenetic mark associated with the repression of gene promoters.

The reference to genome stability was removed due to space constraints.

2. "Its pattern in the genome is disrupted with age and these changes can be used to statistically predict age with epigenetic clocks." Why is the term "disrupted" used here? They just change with age, and we don't know yet if such changes have only negative impact. Some of them can be adaptive, and therefore useful.

By disrupted we mean drastically altered from the developmentally established DNA methylation patterns. The sentence refers to DNA methylation patterns rather than cellular phenotypes.

3. "Rates of aging inferred from these clocks correlate with human health." What is correlation with "health"? If some associations between accelerated age and disease are meant, it should be stated appropriately.

We have amended this sentence:

Altered rates of aging inferred from these clocks are observed in human disease..

4. "However, the molecular mechanisms underpinning age associated DNA methylation changes are unknown." It suggests that mechanisms will be investigated but the study is totally descriptive.

This statement provides background to the study. Our study is conducted using samples from people and therefore is by nature observational. While we cannot directly dissect molecular mechanisms from these observations, our delineation of the role of CpG density in age-associated epigenetic changes is informative when considering the mechanisms underpinning these changes.

5. "Local DNA sequence plays a strong role in programming DNA methylation levels at individual loci independently of age". It makes an impression that the age-dependent impact of DNA sequence [variation] on DNA methylation levels has been investigated but, in fact, this question is quite new. The paper would benefit significantly if the key question is better explained.

We have altered this sentence to:

Local DNA sequence can program steady-state DNA methylation levels.

Page 4.

The decision to limit the study to individuals with 3 datapoints, discarding more than 400 samples, seems to be a counterproductive given that the key question was how DNA sequence variation contributes to aging trajectories of DNA methylation. The larger the sample, the higher the power, the more robust results.

We limited our study to individuals with 3 or more datapoints because estimates of individual slopes would be less robust for individuals with 2 datapoints and impossible for individuals with 1 data point. A mixed effects model provides an alternative approach to include all individuals (discussed in more detail below in response to reviewer 3). However, missing data points in a mixed effects model should satisfy the condition of being missing at random within the dataset (ie randomly distributed across time points). This is not the case in the Lothian Birth Cohort as missing observations are more likely to be from later timepoints due to the earlier death of some individuals. This violation of assumptions prevents us applying such a model to increase the sample size of our analysis.

Lines 28-47. Also not clear why the detected changes were compared to the cytosines from the epigenetic clock. What does it say about the reliability of measurements? The Horvath clock is generated of maximally informative cytosines which do not necessarily strongly correlate with age.

We examined the epigenetic clock CpGs because they are loci whose change in DNA methylation with age has been extensively validated. While the reviewer is correct that they are not necessarily the loci with the strongest age correlations, their use as clocks means that they have been shown to reproducibly change in DNA methylation levels with age in multiple, independent cohorts. We therefore asked whether we could observe their change in methylation using our approach. We have added further discussion of our reasons for analysing epigenetic clock loci to the revised manuscript:

To further test whether we could reliably estimate changes at individual loci, we next examined epigenetic clock loci. Epigenetic clocks have been defined as statistical instruments whose output is strongly correlated with age (20). Individual clock loci show DNA methylation levels that correlate with age in cross-sectional analyses (12). Given their use in predicting age, they would be expected to show consistent trajectories between individuals.

Page 5.

1.Lines 21-25. Correction for blood cell count differences - was it done just in this one instance as a "check" and the resulting figure is added to the supplement (Sup Fig 2 b). ? If it was not done before any downstream analysis, age related cellular effects can be a major confounder. This could be the main reason for inconsistencies with previous epigenetic aging studies, including the predominant age-dependent gain of DNA methylation marks (lines 9-10). This is a major problem with this study which requires a full re-analysis.

We did not control for cell type proportions in our initial analyses because DNA methylation changes occurring within the age range of 67-80 remain under characterised. We therefore wanted to quantify the degree to which variation in the proportions of cell types might cause observed changes in DNA methylation to compare with previous studies rather than directly exclude them before analysis.

The original manuscript included analyses of rapid gain CpGs (now supplementary figure 2c) and the association between CpG density and rate of change of DNA methylation (supplementary figure 4c) after correction for cell counts. In the revised manuscript we have also added an analysis of the proportion of significantly changing CpGs which might be explained by variation in the proportions of cell types (detailed in response to reviewer 3) and an analysis showing that the association between slope QTLs lead SNPs and CpGs remain significant after correction for measured cell counts (supplementary table 3).

2. Continuing with the theme of cell count correction - according to the methods all the lymphocytes were grouped under one cell type. The most popular cell-count correction method for this type of data (Illumina 450K arrays) - the Houseman's algorithm with Reinius et.al. dataset as a reference - includes 4 separate lymphocyte fractions - CD4 T-cells, CD8 T-cells, B-cells and Natural Killers - all accounted for independently. Again, what is the reason for deviating from such standard practice and grouping all these cell types together?

While most studies do estimate cell proportions using DNA methylation data, they lack direct measures of different blood cell types. Direct measures of white blood cell counts are available for LBC as collected on a Sysmex HST system within a UK National Health Service haematology laboratory. These direct measures have the advantage of being entirely independent of the DNA methylation data and thus are unaffected by age associated changes in DNA methylation within a specific cell type (eg Shchukina et al Nature Aging 2021, doi: 10.1038/s43587-020-00002-6). We also note that although these provide less resolution for the different lymphocyte subtypes than the Houseman estimates, they provide greater resolution for granulocytes cell types (neutrophils, basophils and eosinophils) which are provided as a single category following the application of the Houseman method.

To ask the degree to which variation in lymphocyte fractions might explain our observations we have included additional analyses using beta values corrected for cell counts estimated using the Houseman approach. These are as follows:

- analysis of CpGs significantly changing with age (included in text).
- analysis of rapid gain CpGs (supplementary figure 2d).
- analysis of slope-QTLs (supplementary table 3).
- analysis of the relationship between CpG density and rate of change of DNA methylation with age (supplementary figure 4d).

3. Illumina 450K DNA methylation arrays are known to contain batch effects, in particular - sentrix ID and sentrix row position have been shown to have an influence on the estimated DNA modification levels. The paper, however, doesn't say anything about these. It is necessary to know if the sentrix IDs were randomised between the first measurement, and the follow up measurements within the longitudinal design. If the first batch of measurements were placed on one set of arrays, and later follow up experiments were interrogated using another set of arrays, then there is a high chance that some of the results are caused by technical variance.

As the reviewer states, batch effects are observed in Illumina Infinium data. In the case of longitudinal LBC dataset, the data were necessarily generated over time and thus batch is confounded with the variable of interest, age. Thus batch correction is likely to remove biological effects (as detailed in Min et al Bioinformatics 2018, DOI: 10.1093/bioinformatics/bty476). This hypothesis is supported by preliminary work in which we observed that batch correction removed the known changes with age seen at epigenetic clock loci.

To ensure that our observations cannot be accounted for by batch effects, we instead replicated our findings in an entirely independent set of 5,101 individuals from Generation Scotland. As detailed below, the revised manuscript includes an analysis of a further independent set of 4,450 individuals from Generation Scotland. We also note that in the case of our analysis of CpG density, these analyses more than double the number of CpG loci assessed from 219,885 non-CGI probes on the Illumina 450k array for LBC to 593,850 on the EPIC array in Generation Scotland. We argue that this validation in 2 independent sets of individuals and across additional loci strongly suggests that our observations are not the result of batch effects.

4. One general question: what was the point of focusing on the rapid gain and ignoring weaker but still significant aging effects? This hardly understandable choice takes out the context of previous aging studies, making it hard to compare current results with what has been known in the field.

We initially focused on the rapid gain CpGs because they were striking in our analysis of the distribution of age associated changes in DNA methylation (*Figure 2a*) and such an effect had not been reported previously. Their analysis led to the observation that they are in regions of low CpG density. Later in the paper we show that the CpG density plays role more generally, including at CpGs which show weaker changes with age (*Figure 4e*).

5. Furthermore, low DNA modification regions that start aging at 67-80 years of age, is quite unusual. It would be necessary to show if this finding can be replicated in a study with donors collected from a broader age interval. In particular - how the aging profiles of the obtained set of CpGs exhibiting rapid DNA modification changes look like in other public datasets.

As suggested by the reviewer, we have expanded our analysis of Generation Scotland to further explore how local CpG density influences age-associated changes in individuals in a broader range of ages. We have also replicated this in an independent group of individuals from Generation Scotland. We include these analyses in a new results section '*Many age-associated changes in DNA methylation are specific to older individuals*' with data included in *Figure 5* and *Supplementary Figure 5*.

Briefly, our results suggest that the epigenetic changes that occur within the age-range covered by LBC differ from those in the younger individuals which make up the bulk of other studies of age-associated DNA methylation changes. Using Generation Scotland, we find that low CpG density loci lose methylation on average in accordance with previous studies (eg Zhou et al 2018 Nature Genetics, DOI: 10.1038/s41588-018-0073-4). In individuals older than 65 we instead find a reproducible association with variation in DNA methylation levels that our analysis of LBC argues is caused by variation in individual methylation trajectories.

6.Line 32. "Large hypomethylated genomic regions termed partially methylated domains..." Hypomethylated compared to what? Are they "hypomethylated" or PMDs?

Partially methylated domains are hypomethylated compared to the developmentally established landscape of pervasive DNA methylation. We have updated the sentence to make it less ambiguous:

... large genomic regions of reduced methylation termed partially methylated domains (PMDs).

Reviewer 3

In this manuscript, Higham et al. investigate age-related changes in CpG methylation genome-wide, and the genetic basis of such changes, using data from 600 individuals from the Lothian Birth cohort. The main findings are (1) a set of CpGs where there are slope-QTLs; ie. a genetic variation influences

the extent to which methylation at a nearby CpG changes with age. (2) that CpGs in regions of low CpG density are particularly prone to change with age.

Currently, we know that DNA methylation can be used to predict chronological age, but we have little-to-no understanding of why? This paper advances our knowledge of this.

My major concern is technical in nature; the authors have not convinced me that their set of CpGs claimed to be associated with age, are indeed associated with age. I am concerned about cell type composition, as well as replication / can these CpGs actually predict age? The value of the entire paper rests on providing strong, positive answers to these questions. I will expand on these critical questions in the next two paragraphs.

We thank the reviewer for their comments, and we hope that the changes have addressed their concerns.

We know that cell type composition change with age. This has led to a criticism of the Hannum clock (Jaffe and Irizarry, Genome Biology, which should really be discussed in the author's manuscript). First, I note that the authors find that ~50% of CpGs they examine, are also significantly associated with age. That seems to be an astronomical number. While it could be true, the magnitude of this number really suggests confounding by cell type. In response to this, the authors have examined if the age effect disappears when you control for white blood cell counts (an imperfect measure of cell type composition). They claim all their CpGs are still significant when you control for WBC. However, when I compare Fig 2a to Supplemental Fig 2b I see a 10x reduction in effect size (beta value). It is really hard for me to interpret this in any other way than the present study is severely affected by cell type composition. Strangely, in my understanding, the QTL analysis does not correct for WBC which should be a minimum (but perhaps I have misunderstood the manuscript).

As suggested by the reviewer, we have now added further analysis which explore the degree to which variation in the proportion of cell types in the blood might explain our results. As discussed above in response to reviewer 2, these analyses were repeated with our directly measured cell counts and cell counts estimated from DNA methylation data using the Houseman algorithm.

We estimated the degree to which changes in the proportions of cell types explain the 182,760 loci at which we find age-associated changes in DNA methylation levels in LBC. These suggest that 9.77% of significantly age-associated CpGs show rates of change that can be partly explained by alterations in the cellular composition of blood by direct measurements. Furthermore, the proportion of variance explained at these CpGs is low (mean $R^2=0.150$). A larger number of CpGs have rates of change significantly associated with the Houseman estimated cell counts (64.60%) but again the proportion of variance explained at these CpGs is modest (mean $R^2=0.197$).

Regarding the effect size between figure 2a and supplementary figure 2c (previously supplementary figure 2b). In the original manuscript, we scaled beta values to an estimated % methylation but neglected to do for supplementary figure 2c. To avoid confusion, we now represent DNA methylation as beta value or corrected beta value in all figures to make them more directly comparable. This makes it clearer that the effect size is similar between figure 2a and supplementary figure 2c.

In addition, as also discussed in response to reviewer 2, we have now included additional analyses of the slope-QTLs after correcting for measured and estimated blood cell counts as requested by the reviewer (supplementary table 3).

We have also referenced the Jaffe and Irizarry study (reference 23) in our manuscript and added discussion of how changes in cell type proportions relate to our results.

From the introduction:

An analysis of age-associated epigenetic changes has suggested that many correlate with variation in the proportions of different cell types in the blood (22). Furthermore, some epigenetic clocks have been suggested to capture changes in the proportions of cell types because they correlate with cell type proportions (8).

From the discussion:

Previous work has suggested that variation in the proportions of cell types present in the blood may underpin many changes in DNA methylation levels with age (22) and some epigenetic clocks are thought to capture similar changes (8). However, another study has suggested that up to 151,537 loci tested display age-associated changes in DNA methylation when controlling for variation in cell type proportions (77). Here, we find that between 9.77% and 64.60% of the 182,760 loci we find show age-associated changes in DNA methylation in LBC can be partly explained by variation in the proportions of cell types depending on whether we use directly measured or estimated cell counts. However, the proportion of variation explained by cell type variation at these CpGs is modest in both cases. While the majority of studies estimate cell type proportions from DNA methylation data, directly measured cell counts have the advantage of not been confounded by changes in cell-type specific DNA methylation patterns with age (eg as recently described in monocytes) (78). Furthermore, we observe that the effects of slope-QTLs and the association with CpG density we observe are not affected by correction for either directly measured or estimated blood cell counts.

Related, I was looking for evidence that the ~185k age-associated CpGs can be replicated or used to predict age or something else will support the notion that these are actually "true" age-associated CpGs. The reason we (the field) really like Horvath's clock is that it has been consistently replicated. There is some minimal effort spent on this in the manuscript, but the effort falls far short of what I think is required. You could look at replication in a different cohort. You could build an age clock using random subsets of these CpGs and validate this random clock in a new cohort (and repeat for many random samples). However, even if it replicates, it could still be cell type composition.

As detailed in response to reviewer 2, we show that our findings can be reproduced in 2 additional independent sets of individuals aged over 65 from Generation Scotland. We include an additional section detailing these analyses that also asks how age-associated changes in DNA methylation in older individuals correspond to those seen at younger ages:

Many age-associated changes in DNA methylation are specific to older individuals

Data from these analyses are included in with data included in *Figure 5* and *Supplementary Figure 5*. We also include discussion of the implications of our findings:

By comparison to Generation Scotland, we find that the changes in methylation we observe in LBC are replicated in individuals aged over 65 but not in younger members of Generation Scotland. In younger individuals we observe that low CpG density regions show an overall loss of DNA methylation but in older individuals low CpG density regions instead show variable trajectories between individuals. These observations are consistent with a non-linear trajectory for low CpG density regions over the life-course. Previous work has shown an overall loss of DNA methylation with age (5). This is particularly prominent at low CpG density intergenic regions (16) consistent with

our observations in people under 65. The nature of the specific changes in DNA methylation that occur with age in older people remains understudied. One previous study of epigenetic clocks found that their rate of change slows down in later life (81). Taken together this suggests that the nature of DNA methylation changes at different regions varies non-linearly and changes at different points in the life course including in later life. The reasons for the difference in the behaviour of low CpG density regions in older individuals remain unclear and warrant further investigation.

Together, these two issues suggests to me that the work is technically flawed. It is not clear to me what the right approach is to satisfy my concerns about these issues; I hope the authors can do so. Having said that, the idea of the paper is quite interesting and worth pursuing.

We hope the clarifications and additional evidence have satisfied the reviewer as to these points.

Aside from these (potential) technical flaws, here is the usual list of issues

** It would be natural to use the Horvath / Hannum clock to predict age in the data in the paper and assess the predictors. I am especially interested in the Hannum predictor which is less widely used. It would be interesting to look at the epigenetic age "acceleration" (ie. the residual) and see if it is consistent or changes between samples from the same individual.*

We agree this would be interesting, however the focus of our manuscript was on understanding the mechanisms that might underpin age-associated DNA methylation changes rather than analysing epigenetic clocks which are likely to sparsely capture heterogenous processes ongoing in cells and tissues. An analysis relating to this suggestion has also been previously published (Marioni et al 2109 The Journals of Gerontology Series A, DOI: 10.1093/gerona/gly060).

** The authors should explicitly provide the regression model of beta value on age that they fit to estimate the mean changes of methylation per year.*

We have now provided this regression model:

DNA methylation trajectories for each CpG in each individual were modelled by fitting linear models of beta value with age for each CpG and individual using R:

$$\text{equation 1: } \beta_{ij} = \alpha_{ij} \text{age}_i + \gamma_{ij}$$

Where β_{ij} is the beta value of CpG j for individual i, age_i is the age of individual i, α_{ij} is the age effect for CpG j in individual i. γ_{ij} is the intercept for CpG j in individual i. This was only done for individuals and CpGs for which ≥ 3 datapoints were present in the processed dataset. Slopes for each individual and CpG were taken from the linear models (α_{ij}). Mean slopes were calculated as the mean of α_{ij} across all N individuals for CpG j:

$$\text{equation 2: } \mu_j = \frac{1}{N} \sum_{i=1}^N \alpha_{ij}$$

** I would encourage the authors to consider mixed effects models to account for the correlation between samples from the same individual. At the minimum, I would like to see some discussion of why they took their approach and not one of many alternatives.*

Our approach accounts for the correlation between samples for the same individual by analysing each individual independently. As requested, we have added a comparison of our approach to a mixed effects model:

We modelled methylation trajectories for each CpG and individual as linear models of the Infinium beta values with age (example shown in Figure 1a). This approach estimates every individual's slope independently and can account for heterogenous groups within the data. Mixed effect models provide an alternative approach(35). However, mixed effects models borrow information between individuals implicitly assuming all individuals belong to a single group. Our mean rates of change derived from individual trajectories were highly correlated with the rates of change estimated from a mixed effects model including a random intercept (Supplementary Figure 1a, Pearson's $R = 0.999$, $p < 2.2 \times 10^{-16}$ for the 345,890 CpGs that could be modelled in this manner).

- It could be very interesting to see if: (a) different individuals differ with regards to their epigenetic age trajectories at the measured time points, similar to the way different individuals can have different trajectories for individual CpGs (b) there are genetic variants associated with such differences in epigenetic age trajectory

We agree that this would be an interesting study to undertake, but as stated above, our focus is on understanding processes that might explain why particular CpGs might show changes in DNA methylation with age rather than on understanding epigenetic clocks which are likely to capture heterogenous age-associated processes. The size of the Lothian Birth Cohort also means that we are likely to be underpowered to analyse the impact of genetic variation on highly derived phenotypes such as ages estimated by epigenetic clocks.

** With respect to the individual CpGs: (a) are there any known TF motifs overlapping CpGs that show age-associated changes? If so, is the expression of these TFs associated (even if the association is weak) with age (e.g. in GTEx)? This is relevant since abnormal methylation can compromise (or sometimes increase) TF binding, which would essentially mimic reduced (or increased) expression level of the TF.*

As suggested, we have analysed the degree to which the slope QTL CpGs overlap the binding sites of 111 TFs defined by ENCODE:

Cis meth-QTLs have previously been shown to be enriched at enhancers and associated with SNPs altering local transcription factor (TF) binding sites (24, 25). We therefore asked whether this might also be the case for slope-QTLs. We compared the locations of slope-QTL CpGs to binding sites defined for 111 TFs in GM12878 cells (43). Only a single TF, NFE2, was significantly enriched at slope-QTL CpGs and this was only observed at 6 loci (Supplementary Table 6). The vast majority of TF binding sites analysed were significantly instead depleted from slope QTLs (90, 84.91%, Supplementary Table 6).

(b) It would be very informative to do the QTL analysis but focusing only on variants within the coding, and nearby regulatory, regions of epigenetic machinery (especially DNA methylation machinery) genes. This becomes particularly relevant given the (plausible) model of disrupted maintenance at CpG-poor regions that the authors put forward at the end

We believe that this type of study would be informative but beyond the scope of the revision period granted as it would require the development of a new approach to analyse the degree to which genetic changes in genes encoding the DNA methylation machinery associates with variation in how DNA methylation trajectories relate to CpG density.

** As the authors mention in the intro, many changes of methylation occur at CpGs present in CGIs bound by PRC2. Therefore, for figures 2d and 4a, they should separately treat CGIs (and CGI shores) for housekeeping genes and genes which are targets of PRC2 (and often expressed in a tissue/developmental-stage-specific manner).*

In the original manuscript we analysed the overlap of our CpG sets with ChromHMM annotations which include demarcation of regions marked by polycomb repressive complexes and did not see any significant enrichment (Figure 2d and 4b).

To further explore whether polycomb marked CGIs might be specifically enriched, we asked whether rapid gain and slope-QTL CpGs are enriched in CGI probes that overlap ChromHMM polycomb annotation from GM12878 cells. We found that only 9 out of 8,322 of the rapid gain probes (0.108%) and 94 of the 1,456 probe in slope-QTLs (6.456%) were located in CGIs and overlapped polycomb annotations from GM12878 cells. In both cases this is a significant depletion compared to the 9.281% of probes observed for all probes in the analysis ($p < 2.2 \times 10^{-16}$ and $p = 0.0001$ for rapid gain and slope -QTL CpGs respectively). This reproduces our findings using ChromHMM annotations alone.

I have not assess data availability as per Genome Biology standards.

The data used in this study are accessible as detailed in the manuscript:

Data availability

According to the terms of consent for Lothian Birth Cohort 1936 data are available on request from the Lothian Birth Cohort Study, University of Edinburgh (simon.cox@ed.ac.uk). Similarly, according to the terms of consent for Generation Scotland participants, access to data must be reviewed by the Generation Scotland Access Committee. Applications should be made to access@generationscotland.org.

Second round of review

Reviewer 1

The authors have done a good job in addressing the reviewers comments. The only major issue that has come up in the second round is that it looks like a significant number of their hits are most likely due to cell composition effects (indeed the additional analysis associating SNPs with previous GWAS results also supports this e.g. SNPs associated with platelet counts). The authors do address this in the discussion, so thats fine.